# Interfacial chemical bond and internal electric field modulated Z-scheme $S_v$-ZnIn$_2$S$_4$/MoSe$_2$ photocatalyst for efficient hydrogen evolution

Xuehua Wang[1], Xianghu Wang[2], Jianfeng Huang[3], Shaoxiang Li[4], Alan Meng[2 ✉] & Zhenjiang Li [1,4,5 ✉]

Construction of Z-scheme heterostructure is of great significance for realizing efficient photocatalytic water splitting. However, the conscious modulation of Z-scheme charge transfer is still a great challenge. Herein, interfacial Mo-S bond and internal electric field modulated Z-scheme heterostructure composed by sulfur vacancies-rich ZnIn$_2$S$_4$ and MoSe$_2$ was rationally fabricated for efficient photocatalytic hydrogen evolution. Systematic investigations reveal that Mo-S bond and internal electric field induce the Z-scheme charge transfer mechanism as confirmed by the surface photovoltage spectra, DMPO spin-trapping electron paramagnetic resonance spectra and density functional theory calculations. Under the intense synergy among the Mo-S bond, internal electric field and S-vacancies, the optimized photocatalyst exhibits high hydrogen evolution rate of 63.21 mmol•g$^{-1}$•h$^{-1}$ with an apparent quantum yield of 76.48% at 420 nm monochromatic light, which is about 18.8-fold of the pristine ZIS. This work affords a useful inspiration on consciously modulating Z-scheme charge transfer by atomic-level interface control and internal electric field to signally promote the photocatalytic performance.

[1] College of Materials Science and Engineering, Qingdao University of Science and Technology, Qingdao, Shandong, P. R. China. [2] Key Laboratory of Optic-electric Sensing and Analytical Chemistry for Life Science, MOE. College of Chemistry and Molecular Engineering, Qingdao University of Science and Technology, Qingdao, Shandong, P. R. China. [3] School of Material Science and Engineering, International S&T Cooperation Foundation of Shaanxi Province, Xi'an Key Laboratory of Green Manufacture of Ceramic Materials, Shaanxi University of Science and Technology, Xi'an, China. [4] Shandong Engineering Technology Research Center for Advanced Coating, Qingdao University of Science and Technology, Qingdao, P. R. China. [5] College of Sino-German Science and Technology, Qingdao University of Science and Technology, Qingdao, Shandong, P. R. China. ✉email: alanmengqust@163.com; zhenjiangli@qust.edu.cn

With the rapid development of the industrial society, the energy crisis and environmental pollution issues are getting more and more serious. Therefore, finding an alternative energy source is of great significance for the long-term development of human society. Hydrogen ($H_2$) has long been considered as an excellent candidate to substitute the fossil fuel, due to its advantages of clean, renewable, high energy density, and transportability[1–3]. However, at present, the low efficiency, high energy consumption, and environmentally hazardous $H_2$ production technology seriously restrict the commercial application of hydrogen energy. By comparison, photocatalytic water splitting can tactfully convert the sustainable solar energy to $H_2$ energy without discharging any pollutant during the whole process, thus has been considered as a sustainable and promising technique[1,4,5].

In the past few years, metal chalcogenide semiconductor photocatalyst, such as ZnS, CdS, PbS, $ZnIn_2S_4$, have absorbed extensively attention due to the favorable visible-light response ability[6–8]. $ZnIn_2S_4$ is a typical ternary layered metal chalcogenide semiconductor with adjustable band gap of 2.06~2.85 eV, besides, the conduction band is about −1.21 eV, suggesting the intense reducing capacity of the photogenerated electrons[9]. In addition to the suitable band structure, $ZnIn_2S_4$ also possess the prominent photo-stability, environmental and human friendliness in comparison to CdS and PbS[10]. Whereas, the photocatalytic property of the single $ZnIn_2S_4$ is unsatisfying because of the serious carrier recombination. For pursuing the higher photocatalytic activity of $ZnIn_2S_4$, researchers have thrown tremendous efforts, including phase and morphology regulating, elements doping, cocatalyst-loading, defect engineering, and heterojunction constructing[11–15]. Among these strategies, defect engineering and heterojunction constructing are the two-effective means. In photocatalytic field, introducing anion vacancies in semiconductor can not only enhance the light absorption ability of the pristine semiconductor, but also introduce mid gap states in the band gap, which can serve as effective electron "traps" accelerating the separation efficiency of photocarriers[16]. Nevertheless, the excessive defects in photocatalyst can also act as the recombination sites of photocarriers, thus deteriorating the photocatalytic performance[17]. Therefore, regulating the defect in an appropriate concentration would ensure the high activity and stability of photocatalyst[18]. In addition, as known from the reported literatures, the only defect introduction is not enough for realizing efficient photocatalytic property.

Heterojunction constructed by coupling different materials with diverse energy level structure is another effective means to improve photocatalytic performance[19–23]. In recent years, Z-scheme heterostructure, especially the direct Z-scheme heterostructure, has become one of the most effective strategy for obtaining high-efficient photocatalyst[22,23]. For example, Huang et al. reported a HxMoO3@ZnIn2S4 direct Z-scheme photocatalyst for efficient hydrogen production. The results demonstrates that the $H_xMoO_3@ZnIn_2S_4$ presents a 10.5 times higher $H_2$-production activity (5.9 mmol·$g^{−1}$·$h^{−1}$) than pristine $ZnIn_2S_4$[24]. To fabricate the Z-scheme heterostructure, the primary premise is the matching band structure, in which the conduction band of one semiconductor should locate as close to the valence band of another semiconductor as possible. It is reported that the conduction band potential of $MoSe_2$ (about −0.45 eV[25]) is lower than the conduction band of $ZnIn_2S_4$, but very close to its valence band (0.99 eV[9]), which suggests that the photogenerated electrons in the conduction band of $MoSe_2$ are likely to recombine with the photogenerated holes in the valence band of $ZnIn_2S_4$ following Z-scheme pathway. However, as known from the current literatures, $MoSe_2$ can only play the role of cocatalyst in $ZnIn_2S_4/MoSe_2$ instead of realizing Z-scheme

charge transfer[26,27]. The question is that there is no direct and intimate interfacial connection between $MoSe_2$ and $ZnIn_2S_4$. The poor interfacial contact is like erecting a "wall" between the two semiconductors, seriously preventing the trajection of charge flow. Therefore, the formation of intimate interface contact became the hinge to Z-scheme photocatalyst fabrication.

Recently, defect-induced heterostructure construction have opened thought for assembling the heterostructure with specific atomic-level interfacial contact[22]. Its basic principle lies on that the defective sites with abundant coordinative unsaturation atoms and delocalize local electrons can act as the anchoring sites for other semiconductors to form a unique heterostructure contact interface with chemical bond connection[28]. The interfacial chemical bond can act as specific "bridge" accelerating charge transfer between semiconductors. In addition to the intimate interface combination, internal electric field also emerging as a viable strategy to promote Z-scheme charge transfer[29]. Under the effect of internal electric field, the photogenerated electrons in the conduction band of one semiconductor with lower Fermi level could directionally transfer to the valence band of another semiconductor with higher Fermi level, thus realizing the Z-scheme charge transfer[30]. Inspired by the above considerations, an efficient Z-scheme photocatalyst can be obtained through establishing intimate interfacial chemical bond connection between two semiconductors with specific band structure and Fermi level. Up to now, however, the interfacial bonding and internal electric field are always considered separately, the jointly modulation and their synergy effect on photocatalytic performance still remains a challenging task.

Herein, taking S vacancies-rich $ZnIn_2S_4$ ($S_v$-ZIS) and $MoSe_2$ as model material, through a defect-induced heterostructure constructing strategy, an interfacial Mo-S bond and internal electric field modulated Z-scheme $S_v$-ZIS/$MoSe_2$ photocatalyst was fabricated. The addition of hydrazine monohydrate ($N_2H_4 \cdot H_2O$) provides pivotal prerequisite for the formation of S vacancies and coordinative unsaturation S atoms, where the S vacancies can enhance light absorption and facilitate photocarriers separation, while the abundant coordinative unsaturation S atoms can serve as anchoring sites for Mo atoms, thus contributing the formation of Mo-S bond and the in-situ growth of $MoSe_2$ on the surface of $S_v$-ZIS (as showing in Fig. 1). During photocatalytic reaction, the internal electric field induced by the different work function between $S_v$-ZIS and $MoSe_2$ provide intense driving force steering the photogenerated electrons on the conduction band of $MoSe_2$ transfer to the valence band of $S_v$-ZIS, that's the Z-scheme mechanism. Meanwhile, the interfacial Mo-S bond afford the fast pathways for charge transfer from $MoSe_2$ to $S_v$-ZIS, thus accelerating the Z-scheme charge transfer process. This work provides a constructive reference for atomic-level interfacial and internal electric field regulating Z-scheme heterostructure for efficient photocatalytic reaction.

## Results and discussion

**Characterizations of as-prepared photocatalysts**. The morphology and microstructure of the as-synthesized ZIS, $MoSe_2$ and $S_v$-ZIS/$MoSe_2$ (the optimized sample) were analyzed by the SEM, TEM and HRTEM characterizations. As observed in Fig. 2a, the basic morphology of ZIS is flower-like hierarchical microsphere composed by plenty of intersecting nanoflakes, which benefits to the exposure of active surface. The TEM image in Fig. 2b further reveals the hierarchical microsphere of ZIS assembled by nanoflakes. Furtherly, as shown in the HRTEM image in Fig. 2c, the clear lattice stripes with interplanar spacing (d) of 0.32 nm can be well indexed to the (102) lattice plane of hexagonal $ZnIn_2S_4$ (JCPDS:65-2023)[9]. Figures S1–S2 are the elements mapping and

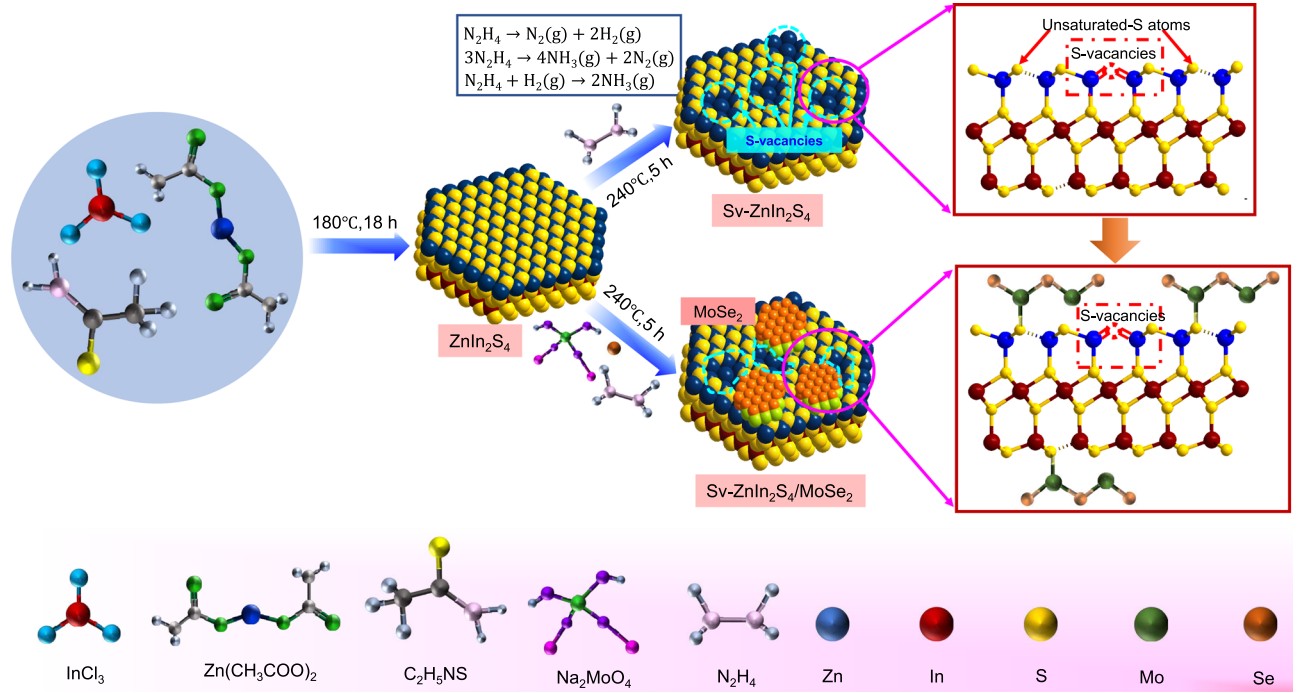

**Fig. 1 Synthesis process.** Schematic presentation of the synthetic route of $S_v$-$ZnIn_2S_4$ and $S_v$-$ZnIn_2S_4$/$MoSe_2$ heterostructure.

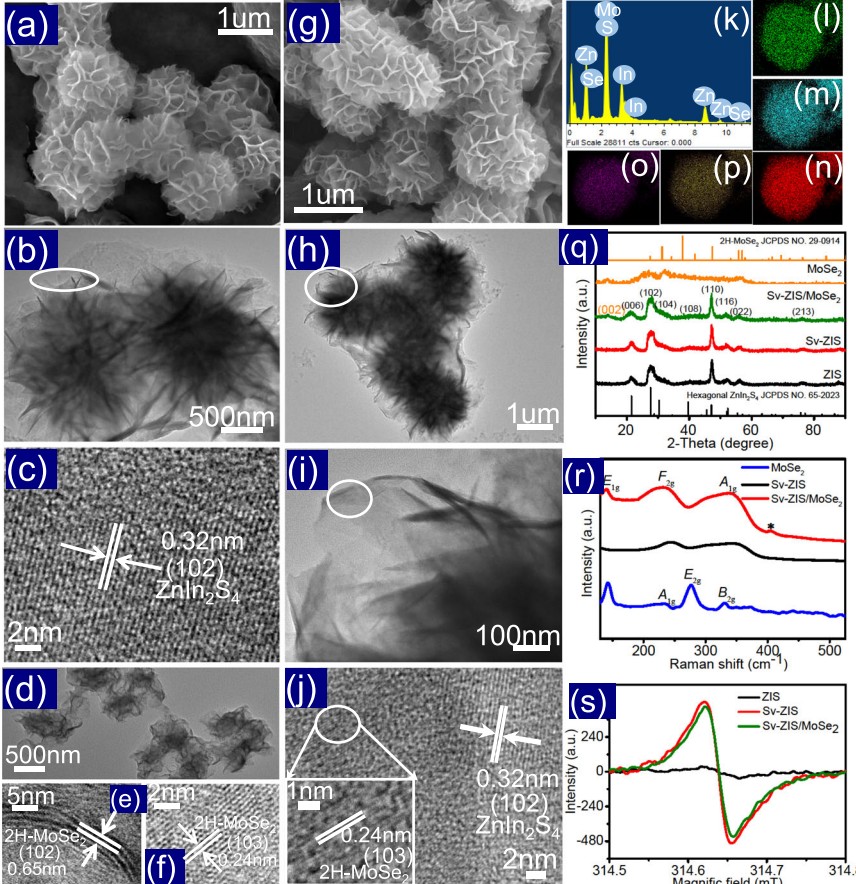

**Fig. 2 Morphology and composition characterizations. a–c** SEM, TEM, and HRTEM pictures of ZIS, **d–f** TEM and HRTEM images of $MoSe_2$, **g–j** SEM, TEM, and HRTEM images of $S_v$-ZIS/$MoSe_2$, **k–p** EDS and elements mapping of Zn, In, S, Mo, and Se in $S_v$-ZIS/$MoSe_2$, **q** XRD patterns of ZIS, $S_v$-ZIS, $MoSe_2$ and $S_v$-ZIS/$MoSe_2$, **r** Raman spectra of $S_v$-ZIS, $MoSe_2$ and $S_v$-ZIS/$MoSe_2$, and **s** EPR spectra of ZIS, $S_v$-ZIS and $S_v$-ZIS/$MoSe_2$.

EDS spectrum of ZIS, it can be clearly seen the evenly distributed Zn, In and S elements, and the atomic ratio of Zn/In/S can be calculated to be about 1.00/1.85/4.13 (as listed in Table S1), very close to the stoichiometric ratio in $ZnIn_2S_4$. Figure S3 presents the SEM, TEM and element mapping images of the $S_v$-ZIS. It is found that the $S_v$-ZIS appears the identical morphology and structure with ZIS, suggesting that the $N_2H_4 \cdot H_2O$-assisted hydrothermal treatment cannot destroy the flower-like microsphere structure of ZIS. The atomic ratio of Zn/In/S in $S_v$-ZIS sample is ~1.00/1.92/3.35 (as displayed in Table S2), the distinctly deficient of S atom compared to that in ZIS confirms the existence of abundant S vacancies in ZIS. Figure 2d is the TEM picture of $MoSe_2$, which manifests the nanosheet feature. The HRTEM image (Fig. 2e and f) present the d-spacing of 0.65 and 0.24 nm, assigning to the (002) and (103) lattice planes of $2H$-$MoSe_2$ (JCPDS: 29-0914), respectively[31]. Figure 2g is the SEM image of $S_v$-ZIS/$MoSe_2$, which exhibits almost the same morphology with ZIS, moreover, the ZIS and $MoSe_2$ in the $S_v$-ZIS/$MoSe_2$ structure are undistinguishable, indicating that the $MoSe_2$ was grown on the surface of ZIS intimately to form a 2D/2D contact, and the introduction of $MoSe_2$ can hardly affect the hierarchical microsphere morphology of ZIS. The TEM image displaying in Fig. 2h and i further reveal the hierarchical flower-like microsphere structure of $S_v$-ZIS/$MoSe_2$, which could lead to the enhanced light absorption by the multilevel reflection and scattering of the incident light[32]. Furthermore, the HRTEM picture displaying in Fig. 2j shows the different lattice stripes with d value of 0.32 and 0.24 nm, respectively, which can be indexed to the (102) crystal face of hexagonal $ZnIn_2S_4$ (JCPDS:65-2023) and the (103) lattice planes of $2H$-$MoSe_2$ (JCPDS: 29-0914), respectively. The HRTEM results indicate that $MoSe_2$ are directly grown and attach on the ZIS nanosheets substrate. Figure 2k-p is the EDS spectra and element mapping of $S_v$-ZIS/$MoSe_2$, as displayed, the distribution of Zn, In, S elements are dense and uniform, meanwhile, the Mo and Se elements are relatively sparse but still evenly distributed. From the EDS spectrum, the mass ratio of $MoSe_2$ to ZIS can be calculated to be about 4.8% (as presented in Table S4), which is very close to the ratio of the added raw materials. What's more, the atomic ratio of Zn/In/S in $S_v$-ZIS/$MoSe_2$ was determined to be 1.00/1.83/3.25, indicating that there is still a mass of S vacancies inside $S_v$-ZIS/$MoSe_2$.

The ZIS, $S_v$-ZIS, $MoSe_2$ and $S_v$-ZIS/$MoSe_2$ were further characterized by X-ray diffraction (XRD) to determine the phase composition. As displayed in Fig. 2q, the XRD pattern of $MoSe_2$ matches well with $2H$-$MoSe_2$ (JCPDS:29-0914)[31]. Meanwhile, ZIS displays the distinct peaks at 21.6°, 27.7°, 30.4°, 39.8°, 47.2°, 52.4°, 55.6° and 76.4°, which can be severally indexed to the (006), (102), (104), (108), (110), (116), (022) and (213) crystal planes of hexagonal $ZnIn_2S_4$ (JCPDS:65-2023)[9]. It is worth noting that the $S_v$-ZIS sample shows almost the same XRD pattern with ZIS, indicating that the introduction of S vacancies can hardly affect the size and crystal structure of ZIS. Moreover, in the XRD patterns of $S_v$-ZIS/$MoSe_2$, in addition to the peaks of hexagonal ZIS, a new peak at about 13.7° can be well assigned to the (002) crystal face of $MoSe_2$, reconfirming the successful synthesis of $S_v$-ZIS/$MoSe_2$ composite.

To further characterize the chemical structures of the as-synthesized photocatalyst, the Raman spectra were carried out (shown in Fig. 2r). As observed in the Raman spectra of $MoSe_2$, the peaks located at 235.4, 277.4 and 330.8 $cm^{-1}$ stem from the $A_{1g}$, $E_{2g}$ and $B_{2g}$ modes of $2H$-$MoSe_2$, respectively, while the peak at 142.1 $cm^{-1}$ is associated to the $E_{1g}$ mode of the in-plane bending of Se atoms in $2H$-$MoSe_2$[32]. For the Raman spectra of $S_v$-ZIS, the peaks located at 244.8 and 348.9 $cm^{-1}$ can be severally assigned to the $F_{2g}$ and $A_{1g}$ modes of $ZnIn_2S_4$. Furtherly, as for the $S_v$-ZIS/$MoSe_2$ (the red line), in addition to

the $E_{1g}$ mode of $2H$-$MoSe_2$, and the $F_{2g}$ and $A_{1g}$ modes of $ZnIn_2S_4$, a new emerging peak situated at about 404.9 $cm^{-1}$ can be indexed to the Mo-S bonding state[33], suggesting that the $S_v$-ZIS and $MoSe_2$ were combined intimately by Mo-S bond. Additionally, it can be observed that all the peaks in $S_v$-ZIS/$MoSe_2$ exhibited evidently blue-shift compared to that in $S_v$-ZIS, further revealing the intense chemical coupling effect between the $S_v$-ZIS and $MoSe_2$[34].

To further testify the existence of S vacancies, the electron paramagnetic resonance (EPR) was carried out (Fig. 2s). For the original ZIS sample, the EPR intensity can hardly be observed, in comparison, the $S_v$-ZIS sample shows the sharply increased EPR signal at a g-factor of 2.009, confirming the abundant S-vacancies in $S_v$-ZIS[35,36]. In addition, it is interesting to observe that the EPR intensity of $S_v$-ZIS/$MoSe_2$ exhibits slightly decreased compared to that of $S_v$-ZIS, which should be contributed to the bonding effect among Mo and unsaturated S in $S_v$-ZIS, decreasing the number of unpaired electrons, but the S vacancies in ZIS have not been sewed up by compositing $MoSe_2$[37].

The X-ray photoelectron spectroscopy (XPS) was applied to investigate the surface composition and chemical states of ZIS, $S_v$-ZIS and $S_v$-ZIS/$MoSe_2$, and the results are showing in Fig. 3. As can be found from the survey spectrum (Fig. 3a), the Zn, In and S peaks are coexisting in ZIS and $S_v$-ZIS, in comparison, Mo and Se peaks can also be observed in the $S_v$-ZIS/$MoSe_2$, which is agree with the EDS test results. As observed in Fig. 3b, the S $2p_{3/2}$ and $2p_{1/2}$ of the original ZIS located at 161.72 and 162.97 eV, respectively, in accordance with the reported literature[36]. In comparison, the S $2p_{3/2}$ and $2p_{1/2}$ of $S_v$-ZIS presented evident negative-shift of about 0.14 eV and 0.19 eV, respectively, verifying the generation of S vacancies in ZIS. The S-vacancies can serve as strong electron-withdrawing group for facilitating the ZIS electrons transfer to S-vacancies, thus decreasing the equilibrium electron cloud density of S atoms inside ZIS, and further leading to the decreased binding energy[38,39]. Furtherly, it can be noted that the S $2p_{3/2}$ and $2p_{1/2}$ of $S_v$-ZIS/$MoSe_2$ exhibited a positive-shift of about 0.13 and 0.17 eV compared to that of $S_v$-ZIS, which should be caused by the strong interfacial interaction between $MoSe_2$ and $S_v$-ZIS[34]. Besides, as shown in Fig. 3c and d, the Zn $2p$ and In $3d$ in $S_v$-ZIS also exhibited a slightly negative-shift compared to that in ZIS, which could be explained that the generation of S vacancies leading to the decreased coordination number of Zn and In[37]. After combining with $MoSe_2$, the Zn $2p$ and In $3d$ peaks re-shift to the high binding energy region, revealing that the bonding effect between Mo atoms in $MoSe_2$ and unsaturated coordination S in $S_v$-ZIS contributing to the slightly increased electron cloud density around Zn and In. Interestingly, it can also be observed that the binding energy variation of Zn $2p$ in ZIS, $S_v$-ZIS, and $S_v$-ZIS/$MoSe_2$ are more notable than that of In $3d$, revealing that the Mo were mainly bonded with the S around Zn sites[37]. What's more, according to the XPS peak area, the actual atomic ratio of Zn/In/S in ZIS, $S_v$-ZIS and $S_v$-ZIS/$MoSe_2$ are 1.00/2.15/3.87, 1.00/2.20/3.29, and 1.00/2.14/3.36, respectively. The lower S atom ratio in $S_v$-ZIS and $S_v$-ZIS/$MoSe_2$ further confirm the presence of abundant S vacancies. As shown in Fig. 3e, the peaks at 228.05 and 230.5 eV can be attributed to Mo $3d_{5/2}$ and $3d_{3/2}$ of $Mo^{4+}$ in $MoSe_2$, meanwhile, the peak at 227.1 eV verified the formation of Mo-S bond[40]. Figure 3f is the Mo $3p$ spectrum, as observed, four distinct XPS peaks can be distinguished, where the peaks at 400.55 and 390.3 eV can be corresponded to the Se Auger peaks, and the peaks at 395 and 416 eV can be assigned to the Mo $3p_{3/2}$ and $3p_{1/2}$ of $Mo^{4+}$. The Se $3d$ spectrum presented in Fig. 3g shows two peaks at 54.4 and 55.35 eV, which can be indexed to Se $3d_{5/2}$ and $3d_{3/2}$ of $Se^{2-}$ in $MoSe_2$, respectively[32]. The XPS results further confirm the successful synthesis of $S_v$-ZIS and

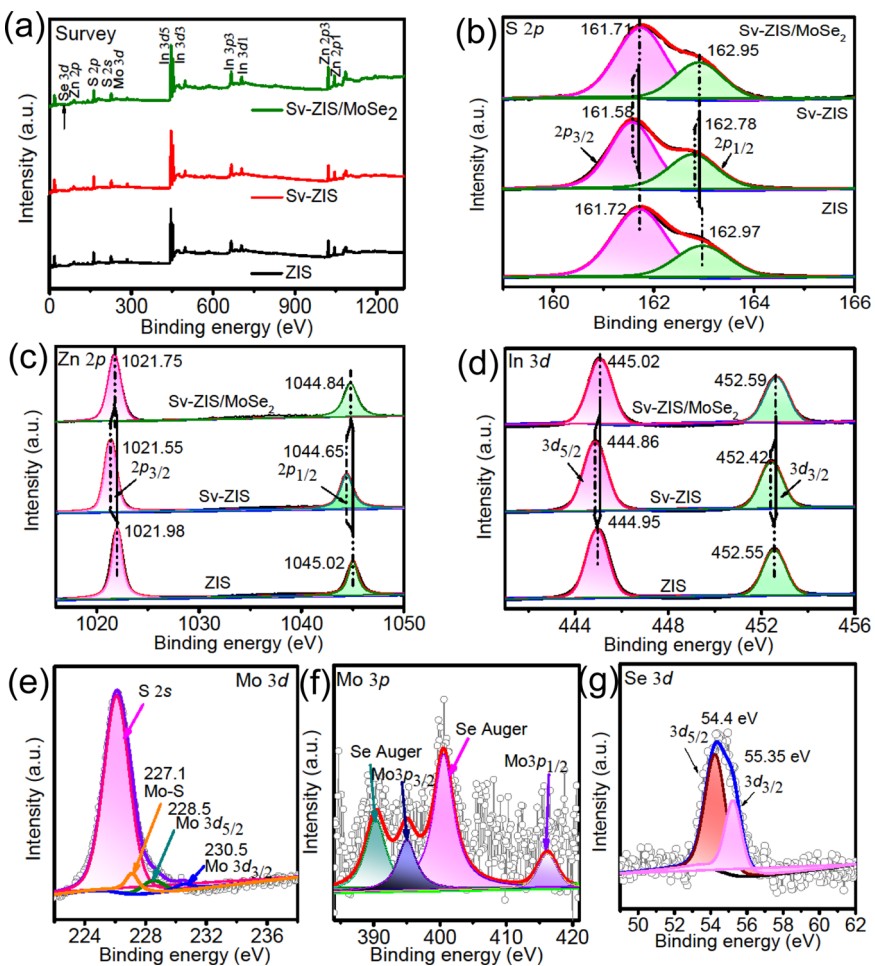

**Fig. 3 XPS spectra. a** survey, **b** S 2*p*, **c** Zn 2*p*, **d** In 3*d* for ZIS, S$_v$-ZIS and S$_v$-ZIS/MoSe$_2$, **e** Mo 3*d* and S 2 *s*, **f** Mo 3*p* and **g** Se 3*d* of S$_v$-ZIS/MoSe$_2$.

S$_v$-ZIS/MoSe$_2$ with abundant S-vacancies, and the MoSe$_2$ is attached on the surface of S$_v$-ZIS through Mo–S bond.

**Photocatalytic H$_2$ evolution activity measurements**. The photocatalytic H$_2$ evolution were evaluated under the visible light ($\lambda > 420$ nm) irradiation, the corresponding test results are showing in Fig. 4. As shown in Fig. 4a and b, all the tested samples exhibit H$_2$ production activity except for MoSe$_2$. The pristine ZIS exhibits the poor H$_2$ production activity of about only 3.36 mmol·g$^{-1}$·h$^{-1}$, in comparison, the S$_v$-ZIS presents a slightly improved H$_2$ evolution rate of 4.77 mmol·g$^{-1}$·h$^{-1}$. The improved photocatalytic performance of S$_v$-ZIS should be ascribed to the accelerated photocarriers separation induced by S vacancies as the electrons trap. Furthermore, the introduction of MoSe$_2$ gave rise to the distinctly improved H$_2$ evolution activity, and the H$_2$ evolution rate of S$_v$-ZIS/MoSe$_2$ increased with the mass ratio of MoSe$_2$ to ZIS increasing. Until the mass ratio of MoSe$_2$ to ZIS reaches to 5.0%, the H$_2$ evolution rate reaches to the highest of 63.21 mmol·g$^{-1}$·h$^{-1}$, which is about 18.8 and 13.3 times higher than that of pristine ZIS and S$_v$-ZIS, respectively, and superior to the recently reported ZnIn$_2$S$_4$-based photocatalytic system (as listed in Table S6). It can also be observed that the S$_v$-ZIS-5.0MoSe$_2$ (synthesized by mixing S$_v$-ZIS and MoSe$_2$ by ultrasound) performs obvious inferior H$_2$ evolution property compared to that of S$_v$-ZIS/5.0MoSe$_2$, indicating that the in-situ growth of MoSe$_2$ on S$_v$-ZIS connecting by Mo–S bond plays critical influence on the photocatalytic performance of the ZIS-MoSe$_2$ composite, which should be attributed to that the Mo–S

bond could facilitate the charge transfer between S$_v$-ZIS and MoSe$_2$. Besides, Fig. S5 shows the wavelength dependent hydrogen evolution efficiency of S$_v$-ZIS/MoSe$_2$, which was tested following the similar procedure of photocatalytic H$_2$ evolution, except that the band-pass filter was equipped to obtain monochromatic incident light ($\lambda$=380, 420, 500 and 600 nm). The detailed test results and the light power of different monochromatic light are displaying in Table S5. Accordingly, the AQY of photocatalytic H$_2$ evolution over the S$_v$-ZIS/MoSe$_2$ photocatalyst can be calculated (the detailed calculation process is shown in the Supporting Information) and the action spectrum was displayed in Fig. 4c. As observed, the action spectrum of S$_v$-ZIS/MoSe$_2$ matches well with the UV-vis absorption spectra, besides, the AQY values of S$_v$-ZIS/MoSe$_2$ are about 93.08% (380 nm), 76.48% (420 nm), 29.7% (500 nm) and 0.15% (600 nm), indicating the favorable optical absorption and utilization capacity of S$_v$-ZIS/ MoSe$_2$ photocatalyst. Fig. S6 is the AQY of ZIS and S$_v$-ZIS, it can be observed that under different monochromatic light wavelength, the AQY of S$_v$-ZIS are larger than that of ZIS, suggesting the more efficient photons to H$_2$ conversion ability of S$_v$-ZIS, which should be caused by the enhanced light absorption and the promoted photocarriers separation efficiency by introducing abundant S-vacancies in S$_v$-ZIS. In addition to the excellent photocatalytic H$_2$ evolution efficiency, the recycling stability is also a pivotal factor for the practical application of photocatalyst. As discerned in Fig. 4d, the H$_2$ evolution amount of the optimized S$_v$-ZIS/MoSe$_2$ photocatalyst remains about 90.5% after 20 h of 5 cycles of photocatalytic tests, signifying the favorable

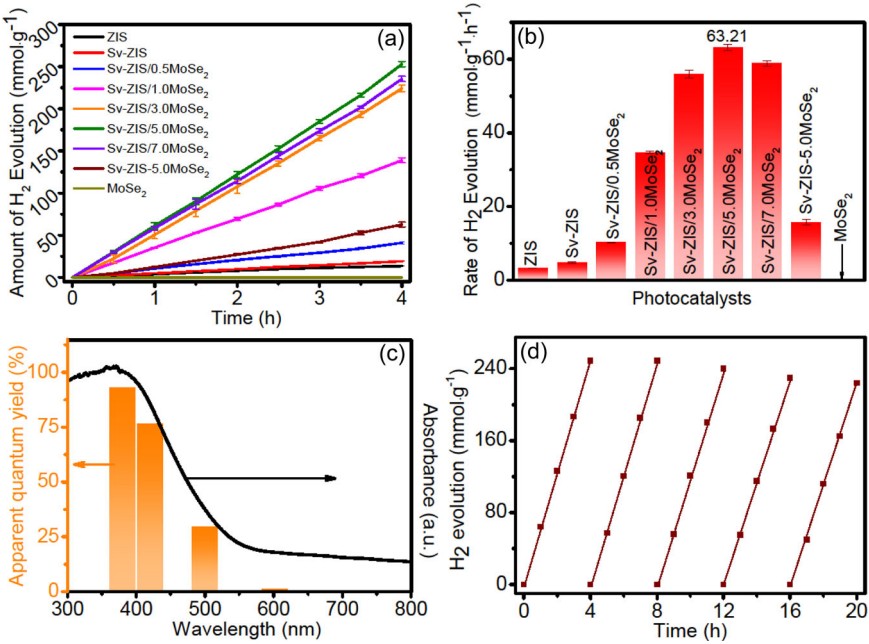

**Fig. 4 Photocatalytic H₂ evolution property. a** H₂ evolution amount at different irradiation time and **b** H₂ evolution rate of different photocatalysts, **c** wavelength-dependent apparent quantum yield (AQY) and **d** cycling stability test of $S_v$-ZIS/5.0MoSe₂. The vertical error bars indicate the maximum and minimum values obtained; the dot represents the average value.

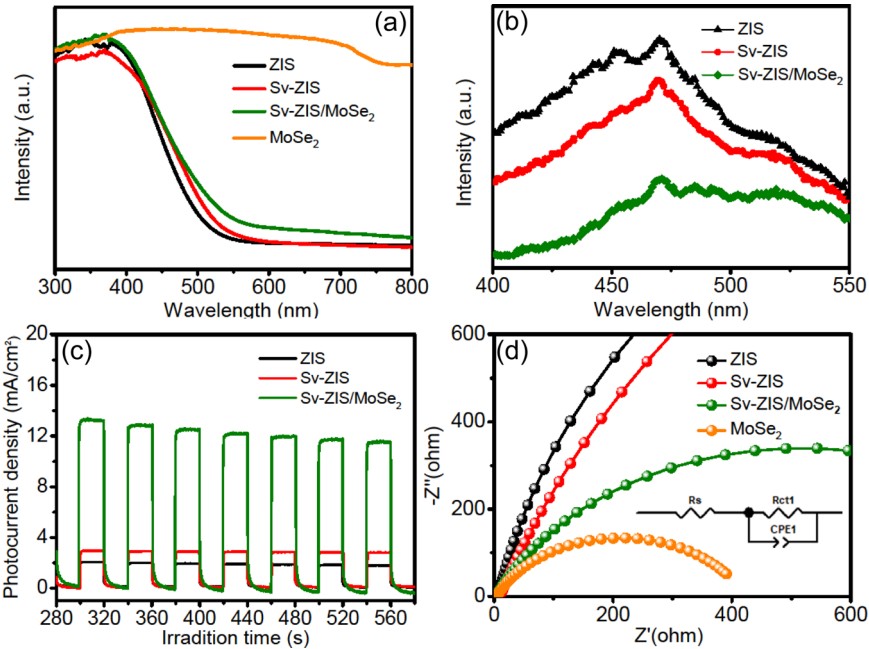

**Fig. 5 Photophysical and Electrochemical measurements. a** UV-vis absorption spectrum, **b** photoluminescence spectra (PL, excited at 375 nm), **c** photocurrent response and **d** electrochemical impedance spectroscopy (EIS) of the as-prepared samples.

photocatalytic stability of $S_v$-ZIS/MoSe₂ photocatalyst, which maybe contributed to the strong combination between ZIS and MoSe₂ through Mo-S bond.

**Photophysical and Electrochemical Properties**. Figure 5a is the UV-vis absorption spectra of ZIS, $S_v$-ZIS, MoSe₂ and $S_v$-ZIS/MoSe₂. It is apparent that the MoSe₂ shows the intense light absorption in the whole UV-vis light range, which should be caused by its dark black color. Meanwhile, it can be observed that light absorption intensity of $S_v$-ZIS is higher than that of ZIS, indicating that the introduction of S vacancies can influence the

band structure of ZIS. Furthermore, after combining with MoSe₂, the light absorption of $S_v$-ZIS/MoSe₂ increased again compared to $S_v$-ZIS. The improved light absorption is in favor of the generation of photocarriers, and beneficial for the enhancement of photocatalytic performance[9]. Figure 5b is the PL spectroscopy. As displayed, under the 375 nm excitation wavelength, the pristine ZIS displays a prominent emission peak, indicating the intense recombination of photogenerated carriers inside ZIS. In comparison to ZIS, the emission peak intensity of $S_v$-ZIS decreases lightly, which should be contributed to that S vacancies can act as electrons trap for facilitating the photocarriers separation. It is

worth noting that the PL signal of $S_v$-ZIS/MoSe$_2$ sample is further quenched compared to that of $S_v$-ZIS, revealing the positive effect of MoSe$_2$ for suppressing the recombination of photocarriers. Figure 5c is the photocurrent response. As observed, all the tested samples exhibit the light-response characteristic under the FX-300 Xe lamp. Obviously, the photocurrent density is in the order of $S_v$-ZIS/MoSe$_2$ > $S_v$-ZIS > ZIS. The highest photocurrent density of $S_v$-ZIS/MoSe$_2$ reveals the most accelerated photocarriers separation and migration efficiency. Figure 5d is the electro-chemical impedance spectroscopy (EIS). As compared, MoSe$_2$ express the smallest semicircle, meanwhile, the semicircle of ZIS is the largest. Obviously, the semicircle of $S_v$-ZIS is slightly lower than that of ZIS, and the semicircle of $S_v$-ZIS/MoSe$_2$ is significantly decreased than that of pristine ZIS and $S_v$-ZIS, manifesting that the introduction of S vacancies and the combination with MoSe$_2$ can decrease the interfacial charge transfer resistance, which is in favor of photogenerated carriers transfer and separation, and finally facilitate the photocatalytic property.

In order to investigate the effects of the MoSe$_2$ to ZIS mass ratio on the photocatalytic performance of $S_v$-ZIS/MoSe$_2$ composites. The light absorption, photocarriers separation and photocurrent density of $S_v$-ZIS/MoSe$_2$ photocatalysts with different mass ratio of MoSe$_2$ to ZIS were also characterized by UV-vis absorption, steady-state PL spectroscopy and photocurrent response. As observed in Fig. S7, with increasing the mass ratio of MoSe$_2$ to ZIS, the light absorption intensity enhance gradually. It is worth mentioning that the $S_v$-ZIS/7.0MoSe$_2$ sample displays the strongest light absorption ability, but its photocatalytic H$_2$ production performance is not the best (as known from Fig. 4a), suggesting that the light absorption is not the only decisive factor for the photocatalytic activity. Fig. S8 is the PL spectra, it can be observed that the PL peak of $S_v$-ZIS/5.0MoSe$_2$ is the lowermost, revealing the most effective photocarriers separation when the mass ratio of MoSe$_2$ to ZIS is 5%, which directly explains why the $S_v$-ZIS/5.0MoSe$_2$ sample has the best photocatalytic performance. Figure S9 shows the photocurrent response. As displayed, the $S_v$-ZIS/5.0MoSe$_2$ shows the highest photocurrent density, which is the result of high-efficiency separation and transfer of photogenerated electron and hole, further revealing the optimum photocatalytic performance of $S_v$-ZIS/5.0MoSe$_2$. As known from the above results, the prominent photocatalytic performance requires the coordination among the efficient light absorption, photocarrier separation and transfer ability.

**Mechanism analysis**. Furthermore, the bandgap value ($E_g$) of the tested sample can be obtained from the Kubelka-Munk function vs. the energy of incident light plots[41]. As displayed in Fig. 6a, the $E_g$ of ZIS, $S_v$-ZIS and $S_v$-ZIS/MoSe$_2$ can be estimated to be 2.35, 2.28 and 2.19 eV, respectively. The narrower $E_g$ is beneficial for the incident light absorption and photocarriers generation, thereby contributing to the photocatalytic property[42]. The Mott-Schottky (M-S) plot can be obtained by the following formula of $C_{sc}^{-2} = \frac{2}{\varepsilon\varepsilon_0 eN_D}\left(E - E_{fb} - \frac{k_B T}{e}\right)$, in which $C_{SC}$ represents space charge capacitance, $\varepsilon$ represents the dielectric constant, $\varepsilon_0$ represents the permittivity of vacuum, e represents the single electron charge, $N_D$ represents the charge carrier density, $E_{fb}$ represents the flat band potential, $k_B$ represents the Boltzmann constant, and T represents the temperature, E represents the electrode potential[9]. As displayed in Fig. 6b-d, the $E_{fb}$ of ZIS, $S_v$-ZIS and MoSe$_2$ can be determined to be −0.96, −0.9 and −0.1 V (vs. NHE), respectively, by extending the linear part of M-S plots. Besides, all the tested samples exhibit the positive slope of M-S plots, indicating the n-type semiconductor traits[43]. As known, the conduction band potential ($E_{CB}$) of n-type semiconductor is

~0.2 eV negative than the $E_{fb}$[44], thus the $E_{CB}$ of ZIS, $S_v$-ZIS and MoSe$_2$ can be discerned to −1.16, −1.1 and −0.3 V (vs. NHE), respectively. According to the equation of $E_{VB} = E_{CB} + E_g$ ($E_{VB}$ is the potential of valence band (VB)), the $E_{VB}$ of the ZIS and $S_v$-ZIS can be estimated to 1.19 and 1.18 V vs. NHE, respectively. According to the reported literature, the $E_g$ of MoSe$_2$ is about 1.89 eV, therefore, the $E_{VB}$ of MoSe$_2$ can be determined to be 1.59 eV[25].

The work function (ϕ) is an important nature for reflecting the escaping ability of free electron from Fermi level ($E_f$) to vacuum level[45]. To investigate the mechanism for the excellent photocatalytic performance of $S_v$-ZIS/MoSe$_2$, the ultraviolet photoelectron spectroscopy (UPS) with He I as the excitation source was conducted. As displayed in Fig. 6e, the secondary cutoff binding energy ($E_{cutoff}$) of $S_v$-ZIS and MoSe$_2$ can be respectively determined as 17.65 and 16.87 eV, by extrapolating the linear part to the base line of the UPS spectra. Based on the formula of ϕ=hv-$E_{cutoff}$, the ϕ of $S_v$-ZIS and MoSe$_2$ can be calculated as 3.57 and 4.35 eV, respectively. Hence, the $E_f$ of $S_v$-ZIS and MoSe$_2$ can be determined as −0.93 and −0.15 V (vs. NHE), respectively. Based on the above calculation and analysis results, the detailed band structure of $S_v$-ZIS, MoSe$_2$ and $S_v$-ZIS/MoSe$_2$ were depicted in Fig. 6f. As observed, the $E_f$ of MoSe$_2$ is below that of $S_v$-ZIS, hence, when $S_v$-ZIS and MoSe$_2$ contact and form an intimate interface, the free electrons in $S_v$-ZIS with high $E_f$ would spontaneously diffuse to MoSe$_2$ with low $E_f$, until a new equilibrium state $E_f$ fabricated. The electron drifting from $S_v$-ZIS to MoSe$_2$ result in the charge redistribution on the interface between $S_v$-ZIS and MoSe$_2$, in which the interface near $S_v$-ZIS side is positively charged, while negatively charged near the MoSe$_2$ side, as result, an internal electric field from $S_v$-ZIS to MoSe$_2$ was built[46].

To further reveal the photocatalytic reaction mechanism of $S_v$-ZnIn$_2$S$_4$/MoSe$_2$ heterostructure, the density functional theory (DFT) calculations were conducted out. Figure 7(a) is the optimized structure of $S_v$-ZnIn$_2$S$_4$/MoSe$_2$ heterostructure, where the coordinative unsaturation S atoms was simulated by breaking two Zn-S bonds in the surface of ZnIn$_2$S$_4$. According to Population analysis and Hirshfeld analysis results, the population of Mo$_{001}$−S$_{018}$ is 0.34, and the transferred charge between MoSe$_2$ and $S_v$-ZnIn$_2$S$_4$ is 0.12|e|. The above results directly demonstrate the intense bonding effect between the Mo atom in MoSe$_2$ and the coordinative unsaturation S atom in ZnIn$_2$S$_4$. Figure 7(b) shows the side view of charge density difference of $S_v$-ZnIn$_2$S$_4$/MoSe$_2$, where the red and blue iso-surfaces denote the accumulation and depletion of electron density, respectively. As observed, the electron cloud density presents distinctly localized distribution between the Mo atom in MoSe$_2$ and the coordinative unsaturation S atoms in $S_v$-ZnIn$_2$S$_4$, which more intuitively manifests the intense bonding effect between Mo and S. In addition, it can be noted that the surface of MoSe$_2$ was dominantly covered by red color, while $S_v$-ZnIn$_2$S$_4$ was chiefly filled by blue color, suggesting that the electrons in $S_v$-ZnIn$_2$S$_4$ were transfer to MoSe$_2$ along the intimate heterointerface, which would subsequently induce the internal electric field in $S_v$-ZnIn$_2$S$_4$/MoSe$_2$ heterostructure[47].

Accordingly, the photocatalytic reaction mechanism of $S_v$-ZIS/MoSe$_2$ can be elaborated in Fig. 7c. Under the irradiation of visible light, a mass of photoinduced electrons (e$^-$) with enough energy would transfer from the VB of $S_v$-ZIS and MoSe$_2$ to the CB of $S_v$-ZIS and MoSe$_2$, respectively, while the holes (h$^+$) be left on the VB of $S_v$-ZIS and MoSe$_2$, respectively. It should be mentioned that the abundant S vacancies inside ZIS could introduce new donor level in the band gap of ZIS, which can act as efficient electrons trap to suppress the photogenerated electron-hole pairs recombination[48]. Furtherly, under the driving

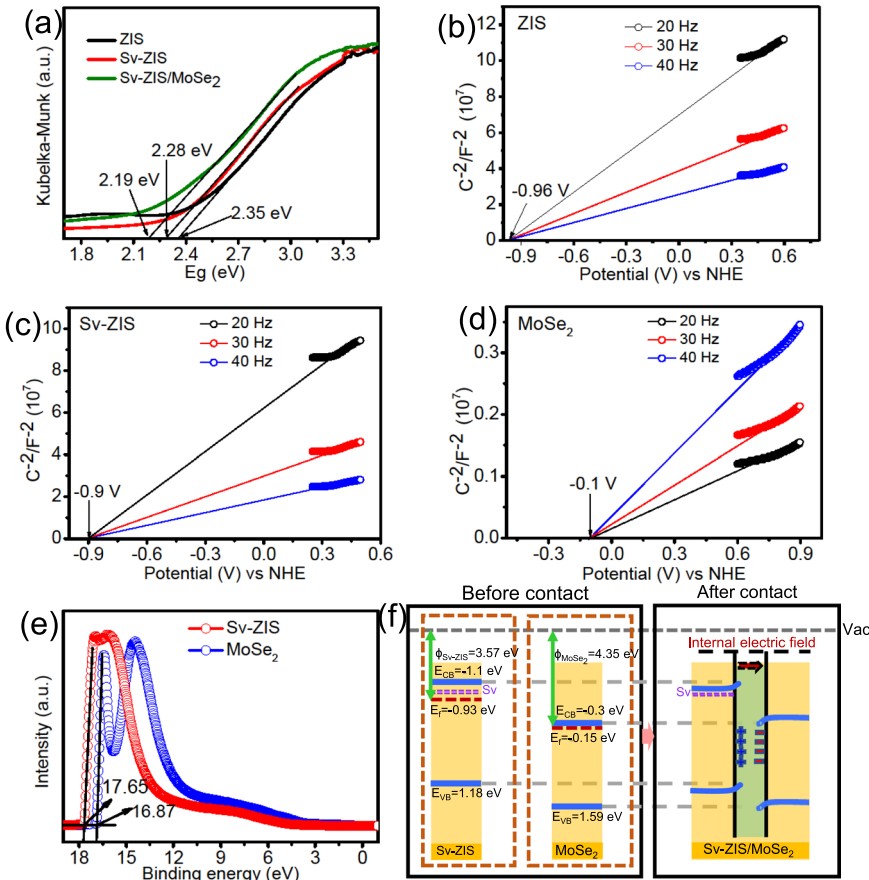

**Fig. 6 Band structure and the formation of internal electric field. a** Kubelka-Munk function vs. the energy of incident light plots, **b–d** Mott-Schottky (M-S) plot, **e** UPS spectra of the as-prepared samples, and **f** band structure of $S_v$-ZIS, $MoSe_2$ and $S_v$-ZIS/$MoSe_2$.

effect of the internal electric field, the electrons on the CB of $MoSe_2$ would migrate to the VB of $S_v$-ZIS to recombine with the holes. The Mo-S bond acting as atomic-level interfacial "bridge" can promote the photoexcited carriers migration between $S_v$-ZIS and $MoSe_2$, thus significantly accelerating the Z-scheme charge transfer. To validate the Z-scheme charge transfer mechanism, the SPV and EPR measurements were carried out. Figure 7d is the SPV spectra of $S_v$-ZIS, $MoSe_2$ and $S_v$-ZIS/$MoSe_2$ samples. It is noted that the pristine $MoSe_2$ presents no SPV signals in the whole wavelength, suggesting the poor photocarriers separation efficiency inside the $MoSe_2$, that's why $MoSe_2$ performed very poor hydrogen evolution. In comparison, a significant positive photovoltage response can be observed in the SPV spectra of $S_v$-ZIS, suggesting that the holes migrate to the surface of $S_v$-ZIS, which is the typical trait of n-type semiconductor[49]. Meanwhile, the SPV response of $S_v$-ZIS/$MoSe_2$ is significantly lower than that of $S_v$-ZIS, which means that fewer photogenerated holes migrate to the surface of $S_v$-ZIS/$MoSe_2$. This phenomenon should be contributed to that the photogenerated electrons on the CB of $MoSe_2$ transfer to the VB of $S_v$-ZIS and recombine with the photogenerated holes, that's the Z-scheme mechanism[50]. EPR spin-trapping experiment with DMPO as spin-trapping reagent was further proceeded to support the Z-scheme charge transfer mechanism in $S_v$-ZIS/$MoSe_2$. As displayed in Fig. 7e, almost no DMPO-·$O_2^-$ signals can be observed under dark conditions. However, under visible light irradiation, the characteristic peaks of DMPO-·$O_2^-$ (1:1:1:1) can be monitored for the $S_v$-ZIS/$MoSe_2$ methanol dispersion liquid, and the peak intensity increase with the time extending, suggesting that the ·$O_2^-$ was generated in the reaction system[51]. In theory, the electrons on $MoSe_2$ cannot

reduce $O_2$ to product ·$O_2^-$ due to the lower CB potential of $MoSe_2$ (−0.3 V vs. NHE) than the redox potential of $O_2$/·$O_2^-$ (−0.33 V vs. NHE)[52]. Therefore, the ·$O_2^-$ should be the reaction product between the photoinduced electrons on the CB of $S_v$-ZIS and $O_2$ (the CB potential of $S_v$-ZIS is about −1.10 eV, lager than the redox potential of $O_2$/·$O_2^-$), indicating that a mass of photogenerated electrons were accumulated on the CB of $S_v$-ZIS under irradiation of visible light, which should be contributed by the recombination between the electron on the CB of $MoSe_2$ and the hole on the VB of $S_v$-ZIS, thus verifying the direct Z-scheme charge migration mechanism. Above SPV and EPR spin-trapping technique provides the direct proof for the direct Z-scheme charge transfer mechanism inside the $S_v$-ZIS/$MoSe_2$ photocatalyst.

In summary, we have successfully demonstrated an interfacial Mo-S bond and internal electric field modulated Z-scheme $S_v$-ZnIn$_2$S$_4$/$MoSe_2$ photocatalyst through a defect-induced heterostructure constructing strategy for boosting the photocatalytic $H_2$ evolution performance. The internal electric field provide the necessary driving force steering the photogenerated electrons on the conduction band of $MoSe_2$ transfer to the valence band of $S_v$-ZnIn$_2$S$_4$ following the Z-scheme mechanism, while the interfacial Mo-S bond creates direct charge transfer channels between $S_v$-ZnIn$_2$S$_4$ and $MoSe_2$, further accelerates the Z-scheme charge transfer process. What's more, the abundant S-vacancies also contribute to the enhanced light absorption and accelerated photocarriers separation. The above factors together lead to the efficient photocatalytic performance of the $S_v$-ZnIn$_2$S$_4$/$MoSe_2$. Specifically, the optimized photocatalyst exhibits a high AQY of 76.48% at 420 nm, and an ~~ultra~~high $H_2$ evolution rate of

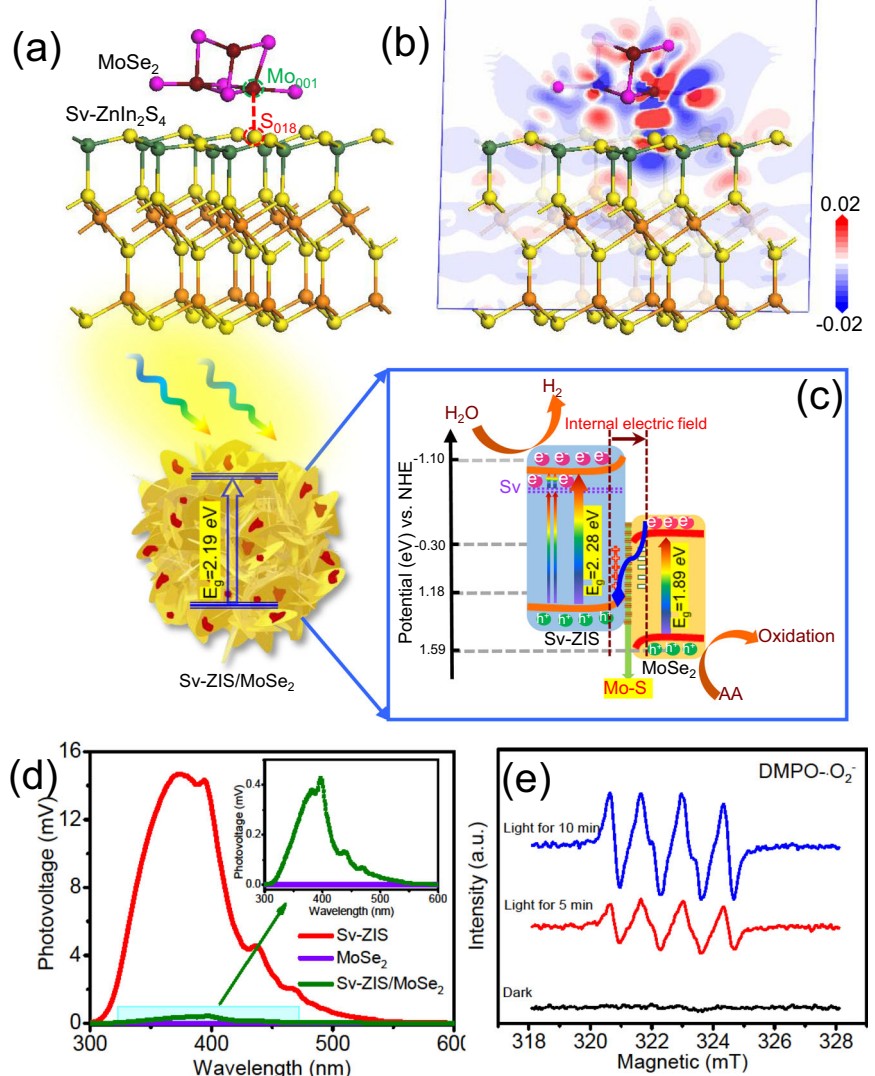

**Fig. 7 Photocatalytic mechanism and verification. a** The optimized structure and **b** the side view of charge density difference of $S_v$-$ZnIn_2S_4$/$MoSe_2$ heterostructure. **c** photocatalytic reaction mechanism of $S_v$-ZIS/$MoSe_2$ under light irradiation, **d** Surface photovoltage (SPV) measurement of $S_v$-ZIS, $MoSe_2$ and $S_v$-ZIS/$MoSe_2$, and **e** DMPO spin-trapping electron paramagnetic resonance (EPR) spectra of DMPO- • $O_2^-$ of $S_v$-ZIS/$MoSe_2$ in methanol solution.

63.21 mmol·g$^{-1}$·h$^{-1}$ under visible light ($\lambda > 420$ nm), which is about 18.8 times higher than that of pristine $ZnIn_2S_4$. Besides, the $S_v$-$ZnIn_2S_4$/$MoSe_2$ also shows favorable recycling stability by remaining above 90% rate retention after 20 h of 5 continuous photocatalytic tests. This work not only provides an efficient direct Z-scheme $ZnIn_2S_4$-based heterostructure photocatalyst, but also affords a beneficial prototype for designing other Z-scheme photocatalyst for efficient green energy conversion.

## Methods

**Materials**. Analytical grade reagents were used directly without purification. Zinc acetate dihydrate ($Zn(CH_3COO)_2 \cdot 2H_2O$) was bought from Tianjin guangcheng chemical reagent Co. LTD. Thioacetamide (TAA), Indium chloride ($InCl_3$), and Selenium power (Se, ≥99.99% metal basis) were bought from Shanghai Macklin biochemical technology Co. LTD. Ascorbic acid (AA) and hydrazine monohydrate ($N_2H_4 \cdot H_2O$, 85%) were bought from Sinopharm Chemical Reagent Co., LTD. Sodium molybdate dihydrate ($Na_2MoO_4 \cdot 2H_2O$) was purchased from Tianjin Fengchuan Chemical Reagent Technology Co., LTD. Deionized water was obtained from local sources.

**Synthesis of $ZnIn_2S_4$ and $S_v$-$ZnIn_2S_4$**. In a representative experiment, $InCl_3$ (1 mmol), $Zn(CH_3COO)_2 \cdot 2H_2O$ (0.5 mmol), and TAA (4 mmol) were orderly dissolved into 50 mL deionized water, and then stirred at room temperature for 30 min. Thereafter, the clear solution was poured into 100 mL stainless steel

autoclave, and maintained at 180 °C oven for 18 h. After cooling naturally to indoor temperature, the sediment was separated by centrifugation, followed by washing with deionized water and ethanol, and drying at 60 °C for 10 h. The obtained yellow powder $ZnIn_2S_4$ were labeled as ZIS. $S_v$-$ZnIn_2S_4$ was prepared via a $N_2H_4 \cdot H_2O$-assisted hydrothermal method. Typically, 100 mg the as-synthesized ZIS was dispersed into 20 mL deionized water for 1 h, then, 5 mL $N_2H_4 \cdot H_2O$ was added into the mixing solution and stirred for another 30 min. After that, the mixture was transfer to 50 mL stainless steel autoclave, and maintained at 240 °C oven for 5 h. Finally, the precipitate was separated by centrifugation, and washing with deionized water for several times, then drying at 60 °C for 10 h. The obtained light-yellow powder was labeled as $S_v$-ZIS.

**Synthesis of $S_v$-$ZnIn_2S_4$/$MoSe_2$ heterostructure**. The $S_v$-$ZnIn_2S_4$/$MoSe_2$ heterostructure were synthesized by the similar process with $S_v$-$ZnIn_2S_4$, except that $Na_2MoO_4 \cdot 2H_2O$ and Se powders were added into the mixture. The $S_v$-$ZnIn_2S_4$/ $MoSe_2$ with different mass ratio of $MoSe_2$ to $ZnIn_2S_4$ (0.5%, 1.0%, 3.0%, 5.0%, and 7.0%) were synthesized by adjusting the addition of $Na_2MoO_4 \cdot 2H_2O$ and Se, and the synthesized samples were labeled as $S_v$-ZIS/0.5$MoSe_2$, $S_v$-ZIS/1.0$MoSe_2$, $S_v$-ZIS/ 3.0$MoSe_2$, $S_v$-ZIS/5.0$MoSe_2$, $S_v$-ZIS/7.0$MoSe_2$, respectively. For comparison, the pure $MoSe_2$ was prepared following the above steps without adding ZIS. Besides, the $S_v$-ZIS-5.0$MoSe_2$ mixture was also fabricated by ultrasonic mixing the $S_v$-ZIS with $MoSe_2$ for 1 h.

**Characterization**. The morphology and microstructure were investigated by SU8010 scanning electron microscope (SEM) outfitted with an energy dispersive

X-ray spectrometer (EDS), and JEM-2100 plus transmission electron microscope (TEM). The crystalline and phase information were characterized by Bruker D8 Advance X-ray diffraction (XRD). The chemical states were investigated by Thermo ESCALAB 250 XI X X-ray photoelectron spectroscopy (XPS, monochromatic Al Kα radiation), and the XPS data was calibrated by C 1 s spectrum (binding energy is 284.8 eV). The light absorption property was researched by the PerkinElmer Lambda 750 S UV-vis spectrophotometer using barium sulfate as standard reference. The recombination of photogenerated carriers was tested by F-4600 spectrofluorometer (375 nm excitation wavelength). The secondary cutoff binding energy was measured by AXIS SUPRA X-ray photoelectron spectroscopy with He I as the excitation source. The surface photovoltage (SPV) measurement were carried out on the system consisting a 500 W Xe lamp source equipped with a monochromator, a lock-in amplifier with a light chopper, a photovoltaic cell, and a computer. The Raman spectra were conducted on LabRAM HR Evolution Raman spectrometer with 325 nm excitation wavelength to analysis the composition. The electron paramagnetic resonance (EPR) measurement was conducted on JEOL JES-FA200 EPR spectrometer with a 9.054 GHz magnetic field. The 5,5-dimethyl-pyrroline N-oxide (DMPO) was adopted as spin-trapping reagent and the $\cdot O_2^-$ and $\cdot OH$ were tested in methanol and aqueous solution, respectively.

**Photocatalytic water splitting for hydrogen evolution**. The hydrogen production experiments were proceeded on Labsolar-6A (Beijing Perfectlight). Typically, photocatalyst (50 mg) was ultrasonically suspended into 100 mL solution involving 0.1 M ascorbic acid sacrificial agent. Prior to exerting light, the reaction system was degassed for 1 h to thoroughly exclude the air and the dissolved oxygen in reaction system. Then the reaction was proceeded under PLS-SEX300D 300 W Xenon lamp (Beijing Perfectlight) with a 420 nm cut-off filter. The light intensity was determined by PLMW2000 photoradiometer (Beijing Perfectlight) to be about 254 mW/cm$^2$. The generated hydrogen was analyzed by GC 7900 gas chromatograph (Techcomp, 5 Å molecular sieve stainless steel packed column, Ar as carrier gas and TCD detector).

**Photoelectrochemical and electrochemical measurements**. All the electrochemical and photoelectrochemical measurements were conducted by a three-electrode system on CHI-660E electrochemical workstation. In the typical three-electrode system, the working electrode was a piece of nickel foam coating with the as-prepared photocatalyst, the reference electrode was Hg/HgO, while the counter electrode was Pt wire. The electrolyte was 0.5 M Na$_2$SO$_4$ aqueous solution. The electrochemical impedance spectroscopy (EIS) was conducted under open-circuit potential with 0.01 to 1×10$^5$ Hz frequency range and 0.005 V AC amplitude. The photocurrent response was tested under FX-300 Xe lamp. Mott-Schottky (M-S) plots were collected from −1 to −0.2 V under 10 kHz frequency and 0.01 V amplitude.

The working electrode was fabricated as follows: a certain amount of photocatalyst, carbon black and polyvinylidene fluoride were weighted according to the mass ratio of 8:1:1, and then dispersed into N-methyl-2-pyrrolidone to gain a homogeneous paste. The paste was daubed on a piece of pre-cleaned 1×1 cm$^2$ FTO collector, and then dried in 60 °C vacuum for 1 h.

**Theoretical calculation**. Density functional theory (DFT) calculations were performed utilizing the CASTEP module of Materials Studio 6.1[53], the Perdew-Burke-Emzerhof (PBE) functional[54], and ultrasoft pseudopotential (USPP) method[55,56]. The cut-off kinetic energy of 400 eV, a 3×3×3 Monkhorst-pack k-point (Γ point) mesh sampled the Brillouin zone with a smearing broadening of 0.05 eV were applied during the whole process. The convergence criteria of self-consistent field (SCF), total energy difference, maximum force, and maximum displacement are 2.0×10$^{-6}$ eV/atom, 2.0×10$^{-5}$ eV/atom, 5.0×10$^{-2}$ eV/Å, and 2.0×10$^{-3}$ Å, respectively.

## Data availability
The experimental data that support the findings of this study are available from the corresponding author upon reasonable request. Source data are provided with this paper.

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

## Acknowledgements

The work reported here was supported by the National Natural Science Foundation of China under Grant No. 51672144, 51572137, 51702181, 52072196, 52002199, 52002200, Major Basic Research Program of Natural Science Foundation of Shandong Province under Grant No. ZR2020ZD09, Shandong Provincial Key Research and Development Program (SPKR&DP) under Grant No. 2019GGX102055, the Natural Science Foundation of Shandong Province under Grant No. ZR2019BEM042, the Innovation and Technology Program of Shandong Province under Grant No. 2020KJA004, Guangdong Basic and Applied Basic Research Foundation (Grant No. 2019A1515110933), China Postdoctoral Science Foundation (Grant No. 2020M683450) and the Taishan Scholars Program of Shandong Province under No. ts201511034. We express our grateful thanks to them for their financial support.

## Author contributions

X.W(1)., A.M., and Z.L. conceived the research. X.W(1). and X.W(2). prepared photocatalysts and conducted all the experiments. X.W(2). and J.H. performed the electrochemistry measurement. S.L. offered help to analyze the characterization experiment data. X.W(1)., X.W(2)., and Z.L. wrote and revised the manuscript. A.M., S.L., and J.H. gave suggestions on the experiment and writing.

## Competing interests

The authors declare no competing interests.
