## [Peer Review File · Nature Communications]

Reviewers' comments:

Reviewer #1 (Remarks to the Author):

The manuscript on defective Z-scheme photocatalyst by X. Wang et al. is interesting and worth to be published after some improvements, as pointed below:

- 1) Introduction could be improved. It is unclear why ZnIn₂S₄ and MoSe₂ have been selected for the construction of Z-scheme photocatalyst. Previous papers on similar systems should be discussed rather than basic/fundamental knowledge on solar photocatalysis.
- 2) Authors have forgotten to discuss the well-known possibility that defects might work as recombination centers for charge carriers (no info about it in Introduction), i.e., although vis response could be obtained by defective materials, they can lose activity under UV irradiation.
- 3) The resolution of EDS is too low for the conclusion on the distribution of elements. It is clear that we can see even elements (Zn, In and S) at the places where there is no sample – the left and right bottom (Fig. S1).
- 4) It is known that for doped samples the shift of XRD peaks is observed. What about self-doped samples (defective)? Please, discuss this with relevant literature.
- 5) Authors have not written how they evaluated XPS data. What kind of reference (if any) was used for peak shift (usually carbon is used).
- 6) Fig. 4c is quite surprising as in experimental part there is no info that band-pass filters have been used (only in SI). Additionally, it is not known what authors mean by "intensity" (photo-absorption properties)?
- 7) I cannot agree that 90% is excellent stability. What is the reason of activity loss (shown both in Figure 4c and 5c).
- 8) For photocatalysis study (in contrast to "dark" catalysis), activity data are usually not shown "per weight" (g⁻¹), as the amount of photocatalyst is not as important as optimal photoabsorption that is reached at ca. 1-2 g/L (see H. Kisch comment in Angew Chem Int Ed). Authors have used 0.5 g/L, and thus it is possible that not at the optimum, which might result in slight differences between samples (e.g., Fig. 4a), but it is not sure that really sample with 5% MoSe₂ is the most active. I think that in future authors could check at which conditions (photocatalyst content), the best activity is obtained.
- 9) There are many unclear statements and mistakes/errors, e.g., "are becoming increasing", "momentous significance", "Z-scheme" vs. "Z-Scheme", "continuously modulate of Z-scheme".
- 10) Additionally, I think that "Vs" term is confusing, as sulfur symbol should be written as "S" (from capital letter). I do recommend changing this into "vS", "Sv" or "Svac."
- 11) Moreover, I wonder if similar research has been performed only in China – only 5 references out of ca. 50 outside China.

Reviewer #2 (Remarks to the Author):

The authors presented Z-scheme heterostructure composed by sulfur vacancies (Vs)-rich ZnIn₂S₄ and MoS₂. In my opinion, the manuscript could be published after major revision. The main points of criticism and question to authors are given below:

1. Sulphur vacancies were created by introduction of N₂H₂·H₂O during ZnIn₂S₄ synthesis, however, only one stoichiometric ration (between N₂H₄·H₂O and other reagents) have been used. Why? How did the authors choose the amount of hydrazine introduced into reaction environment?
2. Photolysis results of sacrificial agent in hydrogen generation should be given
3. In figure 6d, 1%C@HPW/CdS; 2%C@HPW/CdS; 3%C@HPW/CdS; 5%C@HPW/CdS and 7%C@HPW/CdS samples are listed (containing different amount of HPW), but these sample are not mentioned in any other part of the manuscript.
4. How the amount of sulphur vacancies could be controlled?

Reviewer #3 (Remarks to the Author):

The authors have synthesized a sulfur vacancy rich ZnIn₂S₄ and MoSe₂ heterostructure through in-situ growth for efficient photocatalytic hydrogen evolution. They have demonstrated that the intimate interfacial connection and Z-scheme charge transfer system between MoSe₂ and ZnIn₂S₄ allows for a markedly improved hydrogen evolution rate. The manuscript contains thorough discussions, along with well-rounded characterization and performance analysis of the synthesized photocatalyst. However, since sulfur vacancies in ZnIn₂S₄ (Appl. Cat. B 2019, 248, 193-201; Phys. Chem. Chem. Phys. 2019, 21, 25484-25494; J. Energy Chem 2021, 58, 397-407) and the formation of its heterojunction with MoSe₂ (CrystEngComm 2021, 10.1039/D0CE01808B; ChemSusChem 2017, 10, 4624-4631) in photocatalytic hydrogen evolution have been widely reported in recent years, I do not recommend this manuscript to be published in high-impact Nature Communications owing to limited novelty, urgency and scientific merits. The hybrid system is not something new and the synthesis strategy is rather not novel (as claimed in comment #1). Despite extensive characterization analysis and superb photocatalytic performance, this work does not provide new scientific knowledge which can merit wide readership in this field. Nevertheless, I believe this manuscript would be suitable for other publications such as Communications Chemistry. To further improve the quality of the paper, some of the comments are listed below.

- 1) In another study done by Feng et al., (CrystEngComm 2021, Constructing 2D/2D Heterojunction of MoSe₂/ZnIn₂S₄ Nanosheets for Enhanced Photocatalytic Hydrogen Evolution, 10.1039/D0CE01808B), their proposed heterojunction system was that of a p-n system despite using a similar in-situ formation strategy of MoSe₂, albeit without the added modification of the S vacancy. Could the authors provide some insights as to why this could be? It is important to have clear and confirmed charge transfer mechanism.
- 2) The authors have shown that although the light absorption is highest with a mass ratio of MoSe₂ to ZIS is 7%, the H₂ production is not the maximum as it has less effective photocarrier separation as compared to the 5% mass ratio. Could the authors comment on the underlying mechanism behind this? Evidence should be supplemented to strengthen the statement.
- 3) From Figure 6a, the band gap of ZIS, Vs-ZIS and Vs-ZIS/MoSe₂ was 2.35, 2.28 and 2.19 eV, respectively. Why do the authors state the decrease in band gap 2.28 to 2.19 eV after the introduction of MoSe₂ when the heterojunction system consists of the individual band gaps of the two components (Vs-ZIS and MoSe₂) as shown in Figure 7a? How do the authors clarify the 2.19 eV band gap for the Vs-ZIS/MoSe₂.
- 4) It is noted that with the presence of S vacancy, the absorption edge is extended to higher wavelength of light. Could the authors include the action spectra analysis in the performance measure of the synthesized catalyst?

Rebuttal letter

We wish to express our sincere appreciation to Dr. Adam Weingarten and reviewers for reviewing our manuscript entitled “**Interfacial Chemical Bond and Internal Electric Field Modulated Z-Scheme Vs-ZnIn₂S₄/MoSe₂ Photocatalyst for Efficient Hydrogen Evolution**” (NCOMMS-21-04662), which has been submitted to Nature Communications on 4th February 2021. Reviewer#1 and 2 give positive opinions to our work, and recommend this work to be accepted for publication after addressing several questions. After the detail study and analysis, we think their comments are of great significance to improve the quality of our manuscript. Moreover, the comments provided by the Reviewer #3 are also have constructive meanings for our manuscript, however, we think that the overall innovation of our manuscript was not felicitously evaluated by Reviewer #3. According to the three reviewers’ comment, we made the following point-by-point response.

Response to Reviewer #1

Comment 1.1

Introduction could be improved. It is unclear why ZnIn₂S₄ and MoSe₂ have been selected for the construction of Z-scheme photocatalyst. Previous papers on similar systems should be discussed rather than basic/fundamental knowledge on solar photocatalysis.

Response:

Many thanks for the reviewer’s careful comment. According to the reviewer’s suggestion, the **Introduction** section of our manuscript has been reedited, the following description has also been added into the revised manuscript on Line 11-21/Page 3 and highlighted:

In the past few years, metal chalcogenide semiconductor photocatalyst, such as ZnS, CdS, PbS, ZnIn₂S₄, have absorbed extensively attention due to the favorable visible-light response ability^{7,9}. ZnIn₂S₄ is a typical ternary layered metal chalcogenide semiconductor with adjustable band gap of 2.06~2.85 eV, besides, the conduction band is about -1.21 eV, suggesting the intense reducing capacity of the

photogenerated electrons¹⁰. In addition to the suitable band structure, ZnIn₂S₄ also possess the prominent photo-stability, environmental and human friendliness in comparison to CdS and PbS¹¹. Whereas, the photocatalytic property of the single ZnIn₂S₄ is unsatisfying because of the serious carrier recombination. For pursuing the higher photocatalytic activity of ZnIn₂S₄, researchers have thrown tremendous efforts, including phase and morphology regulating, elements doping, cocatalyst-loading, defect engineering and heterojunction constructing¹²⁻¹⁶.

The following description has also been added into the revised manuscript on Line 13-16/Page 4 and highlighted:

For example, Huang et al. reported a H_xMoO₃@ZnIn₂S₄ direct Z-scheme photocatalyst for efficient hydrogen production. The results demonstrates that the H_xMoO₃@ZnIn₂S₄ presents a 10.5 times higher H₂-production activity (5.9 mmol·g⁻¹·h⁻¹) than pristine ZnIn₂S₄²⁵.

The following description has also been added into the revised manuscript on Line 18-22/Page 4 and highlighted:

It is reported that the conduction band potential of MoSe₂ (about -0.45 eV²⁶) is lower than the conduction band of ZnIn₂S₄, but very close to its valence band (0.99 eV¹⁰), which suggests that the photogenerated electrons on the conduction band of MoSe₂ are likely to recombine with the photogenerated holes on the valence band of ZnIn₂S₄ following Z-scheme pathway.

The following references have been listed in the revision, and the order of the entire reference section is rearranged.

7. Kageshima, Y, et al, Photocatalytic and photoelectrochemical hydrogen evolution from water over Cu₂Sn_xGe_{1-x}S₃ particles. *J. Am. Chem. Soc.* (2021). <https://doi.org/10.1021/jacs.0c12140>.
8. Wang, S. B. et al. Supporting Ultrathin ZnIn₂S₄ Nanosheets on Co/N□Doped Graphitic Carbon Nanocages for Efficient Photocatalytic H₂ Generation, *Adv. Mater.* **31**, 1903404 (2019).
9. Kim, Y. et al. Efficient photocatalytic production of hydrogen by exploiting the polydopamine-semiconductor interface, *Appl. Catal. B: Environ.* **280**, 119423 (2021).

10. Li, Z. J. et al. CoNi Bimetal Cocatalyst Modifying a Hierarchical ZnIn₂S₄ Nanosheet-Based Microsphere Noble-Metal-Free Photocatalyst for Efficient Visible-Light-Driven Photocatalytic Hydrogen Production. *ACS Sustain. Chem. Eng.* **7**, 20190-20201 (2019).
11. Mohanty, B. Chattopadhyay, A. Nayak, J. Band gap engineering and enhancement of electrical conductivity in hydrothermally synthesized CeO₂-PbS nanocomposites for solar cell applications. *J. Alloys Compd.* **850**, 156735 (2021).
25. Xing, F. S. et al. Tunable charge transfer efficiency in HxMoO₃@ZnIn₂S₄ hierarchical direct Z-scheme heterojunction toward efficient visible-light-driven hydrogen evolution. *Appl. Catal. B Environ.* **285** 119818 (2021).

Comment 1.2

Authors have forgotten to discuss the well-known possibility that defects might work as recombination centers for charge carriers (no info about it in Introduction), i.e., although vis response could be obtain by defective materials, they can lose activity under UV irradiation.

Response:

Thanks for the reviewer's professional suggestion. According to the reviewer's suggestion, we have added the following discussion about the influence of defects on photocatalytic activity into the **Introduction** section of our manuscript, which has displayed on Line 3-6/Page 4 and highlighted in the revised manuscript:

Nevertheless, the excessive defects formation in photocatalyst can act as the recombination sites of photocarriers, thus deteriorating the photocatalytic performance¹⁸. Therefore, regulating the defect in an appropriate concentration would ensure the high activity and stability of photocatalyst¹⁹.

The following references have also been listed in the revision, and the order of the entire reference section is rearranged.

18. Zhang, Y. et al. Structure-Activity Relationship of Defective Metal-Based Photocatalysts for Water Splitting: Experimental and Theoretical Perspectives. *Adv. Sci.* 1900053 (2019).
19. Liu, J. et al. Defects engineering in photocatalytic water splitting materials. *ChemCatChem.* **11**, 6177-6189 (2019).

Comment 1.3

The resolution of EDS is too low for the conclusion on the distribution of elements. It is clear that we can see even elements (Zn, In and S) at the places where there is no sample-the left and right bottom (Fig. S1).

Response:

Thanks for the reviewer's careful suggestion. In fact, in the original manuscript, due to our negligence, the select of EDS analysis viewshed was inappropriate, which was carried out aiming at the local region in ZnIn_2S_4 flower-like microsphere, thus leading to the full-screen element point distribution. According to the reviewer's suggestion, the EDS in the revised manuscript has been retested, and the results are showing in **Fig. R1**. As observed, the distribution of Zn, In and S elements are well consistent with the morphology of ZnIn_2S_4 . Thanks again for the reviewer's careful comment.

Fig. R1. Morphology characterizations of ZIS. (a) SEM, (b-d) the corresponding element mapping images of Zn, In and S in ZIS.

Fig. R1 has been added into the Supporting Information of the revised

manuscript to replace the original Fig. S1.

Comment 1.4

It is known that for doped samples the shift of XRD peaks is observed. What about self-doped samples (defective)? Please, discuss this with relevant literature.

Response:

Many thanks for the reviewer's professional comment. In response to this issue, we did a very careful literature research, and found that XRD peak shift can be affected by multifarious factors including doping amount, doping type (interstitial- and substitutional-doping), or the size difference between the doped atom (or ion) and ontological atom (or ion).

In detail, for heteroatom doping, the peak shifting usually occurs. For example, Ran et al. synthesized a P-doped graphitic carbon nitride (PCN) nanosheets for visible-light photocatalytic H₂ production. The XRD pattern suggests that the (002) peak of the P-doped sample exhibits an obvious shift to the low diffraction angle compared with the undoped sample, suggesting that the interlayer distance of CN increased after P doping due to the larger radius of the doped P atom than that of the replaced C atom (*Energy Environ. Sci.*, 2015, 8, 3708-3717). Shi et al. reported an interstitial P-doped CdS for photocatalytic water splitting. The XRD results show that the peaks position of CdS-P shift slightly toward the lower diffraction angle, meaning an increased interplanar spacing induced by P-doping (*Adv. Mater.* 2018, 30, 1705941). However, in some cases, it is difficult to observed the distinct XRD peak shift in heteroatom doping. For example, Yang et al. prepared an oxygen-doped ZnIn₂S₄ photocatalyst, which exhibits almost the same XRD peak location with the undoped ZnIn₂S₄ (*Angew. Chem. Int. Ed.* 2016, 55, 6716 -6720).

For self-doped sample, although no foreign atom was introduced, the peak shift can also take place. For instance, Liu et al. reported a Ti³⁺ self-doped TiO₂ microsphere for photocatalytic gas-phase benzene oxidation. The XRD results illustrates that the (101) peaks of Ti³⁺ self-doped TiO₂ sample shows distinct shift to the lower angles compared to the original TiO₂ sample, which should be caused by the larger Ti³⁺ ions (0.067 nm) than that of Ti⁴⁺ ions (0.060 nm) (*Chemical Engineering Journal* 402 (2020) 126220). Huang et al. synthesized a carbon self-doped g-C₃N₄ (CNU) for visible-light-driven hydrogen evolution. XRD result demonstrates that the (002) peaks of the CNU samples are slightly blue-shifted to

27.06° than the pristine g-C₃N₄ (27.47°), revealing the increasing of adjacent-layer gap induced by carbon self-doping (*Applied Catalysis B: Environmental* **254** (2019) **128-134**). Shi et al. reported a self-doped and anisotropy-strengthened Sn_{0.98}Se as exceptional thermoelectric materials. The XRD results illustrates that all the self-doped samples exhibit peak deviation to some extent compared to the standard sample (*Energy Storage Materials* **10** (2018) **130-138**). In addition, there are still some literatures demonstrated that no distinct XRD peak shift can be found in the self-doping sample. For example, Yu et al. synthesized a Ta⁴⁺ self-doped Ta₂O₅/g-C₃N₄ for photocatalytic hydrogen production and RhB degradation. XRD results show that the peaks of Ta⁴⁺ self-doped Ta₂O₅ are almost consistent with the original TiO₂, and no distinct peak shift can be found (*Applied Catalysis B: Environmental* **205** (2017) **271-280**). Zhou et al. reported a Ti³⁺ self-doped mesoporous black TiO₂/graphene assemblies for photocatalytic hydrogen evolution. XRD results demonstrated that no obvious peak shift in the XRD pattern of Ti³⁺ self-doped black TiO₂ can be observed in comparison to the pristine TiO₂ (*Journal of Colloid and Interface Science* **505** (2017) **1031-1038**).

As known for the above literatures, XRD peak shift may occur regardless of heteroatomic or homoatomic doping, which could be influenced by the doping amount, doping types or the size difference between the doped atom (or ion) and ontological atom (or ion). The high doping concentration, large size difference, or interstitial doping usually result in XRD peak shift, but this is not always the case.

In our S_v-ZnIn₂S₄/MoSe₂ system, the addition of highly-reductive hydrazine monohydrate (N₂H₄·H₂O) could release a mass of reductive gases (such as, N₂, H₂) for attacking the Zn-S bond directly. In this case, the S atoms in S_v-ZnIn₂S₄ would become unsaturated, even escape from the lattice of ZnIn₂S₄ and produce S-vacancies. Although the generation of S-vacancies could give rise to the lattice contraction of ZnIn₂S₄, the very low content of S-vacancies in S_v-ZnIn₂S₄ and S_v-ZnIn₂S₄/MoSe₂ lead to the basically unchanged XRD peak location with the pristine ZnIn₂S₄, which are consistent with the reported literatures (*Applied Catalysis B: Environmental* **291** (2021) **120069**; *Adv. Mater.* **2020**, **1907444**; *Nano Energy* **81** (2021) **105613**). Many thanks for the reviewer providing us this enlightening question.

Comment 1.5

Authors have not written how they evaluated XPS data. What kind of reference (if

any) was used for peak shift (usually carbon is used).

Response:

Thanks for the reviewer's careful comment. In our manuscript, the XPS data was calibrated by C1s spectrum (binding energy is 284.8 eV). According to the reviewer's suggestion, the description of XPS in the **Characterizations** section was reedited as follows and highlighted in the revised manuscript on Line 3-5/Page 27.

The chemical states were investigated by Thermo ESCALAB 250 XI X X-ray photoelectron spectroscopy (XPS, monochromatic Al K α radiation), and the XPS data was calibrated by C 1s spectrum (binding energy is 284.8 eV).

Comment 1.6

Fig. 4c is quite surprising as in experimental part there is no info that band-pass filters have been used (only in SI). Additionally, it is not known what authors mean by "intensity" (photo-absorption properties)?

Response:

Thanks for this careful comment. In order to illustrate more clearly, we have added the following description into the revised manuscript on Line 16-23/Page 15 and highlighted:

Besides, Fig. S5 shows the wavelength dependent hydrogen evolution efficiency of S_v-ZIS/MoSe₂, which was tested following the similar procedure of photocatalytic H₂ evolution, except that the band-pass filter was equipped to obtain monochromatic incident light ($\lambda = 380, 420, 500$ and 600 nm). The detailed test results and the light power of different monochromatic light are displaying in Table S5. Accordingly, the AQY of photocatalytic H₂ evolution over the S_v-ZIS/MoSe₂ photocatalyst can be calculated (the detailed calculation process is shown in the Supporting Information) and the action spectrum was displayed in Fig. 4c.

In addition, the "intensity" in the right axis of Fig. 4c in our manuscript means the "absorbance" of the optimized S_v-ZIS/MoSe₂ photocatalyst. To avoid misunderstandings, the "Intensity" in Fig. 4c, Fig. 5a, Fig. S5, S6 and Fig. S7 are replaced by "Absorbance". Thanks again for the reviewer's valuable suggestion.

Comment 1.7

I cannot agree that 90% is excellent stability. What is the reason of activity loss (shown both in Figure 4c and 5c).

Response:

Thanks for the reviewer's careful comment. Just as the reviewer mentioned, a H₂ production retention rate of 90% (Figure 4c) is not excellent enough for the practical production applications in future. Therefore, in the revised manuscript, we changed the description of the cycling stability from "excellent" to "favorable".

As for the reason of activity loss, in our opinion, the following reasons can explain it. Firstly, in the synthesized S_v-ZnIn₂S₄/MoSe₂, the charge transfer follows the Z-scheme mechanism, that is, the photogenerated electrons in the conduction band of MoSe₂ transfer to the valence band of S_v-ZnIn₂S₄ to recombine with photogenerated holes here, while the photoinduced holes with high oxidation capacity left in the valence band of MoSe₂. Although the ascorbic acid sacrificial agent can react with a large portion of photogenerated holes in MoSe₂, there are still some photogenerated holes react with the Se²⁻ in MoSe₂ to generate Se⁰, thus leading to the self-decomposition of MoSe₂, and finally deteriorating the photocatalytic property. In addition, the active site blocking is another factor leading to the decreased photocatalytic H₂ evolution rate. In our work, the magnetic stirring was placed under the reactor to avoid photocatalyst sedimentation. Even so, the agglomeration of photocatalyst is inevitable during long-term photocatalytic reaction. As aggregation occurs, the active sites on photocatalyst would be occluded. Moreover, the constant mechanical force would give rise to the slightly microstructure collapse of S_v-ZnIn₂S₄/MoSe₂, further decreasing the exposure of active sites and the multilevel reflection and scattering of incident light, and finally leading to the deterioration of photocatalytic H₂ evolution performance. Thirdly, in our work, the stability test was carried by replenishing the ascorbic acid sacrificial agent into the reaction solution. After every cycle, the reaction solution was transferred out from the reactor to a beaker, which was subsequently ultrasonic to ensure the sufficiently dissolve of the newly added ascorbic acid. In the process of several transfer of the reaction solution, the loss of photocatalyst in the reaction system is inevitable, which further exacerbates the H₂ evolution property decrease of S_v-ZnIn₂S₄/MoSe₂ photocatalyst. Therefore, in our work, the decrease of photocatalytic reaction stability may be the

combined result of the above-mentioned factors.

Besides, during the photocurrent test (Figure 5c), the working electrode was fabricated by smearing the paste containing the photocatalyst, carbon black and polyvinylidene fluoride on the FTO collector. Due to the relatively weak combination between photocatalyst and FTO glass, the dissolution and shedding of photocatalyst from the FTO glass would be inevitable, thus leading to the slightly decrease of photocurrent density. Therefore, the decreased photocurrent density may be mainly due to the instability of the photoelectrode, rather than the instability of photocatalyst. Thanks again for this enlightening comment.

Comment 1.8

For photocatalysis study (in contrast to “dark” catalysis), activity data are usually not shown “per weight” (g^{-1}), as the amount of photocatalyst is not as important as optimal photoabsorption that is reached at ca. 1-2 g/L (see H. Kisch comment in *Angew Chem Int Ed*). Authors have used 0.5 g/L, and thus it is possible that not at the optimum, which might result in slight differences between samples (e.g., Fig. 4a), but it is not sure that really sample with 5% MoSe₂ is the most active. I think that in future authors could check at which conditions (photocatalyst content), the best activity is obtained.

Response:

Many thanks for the reviewer’s professional comment. As for the activity data expression in photocatalytic H₂ evolution, there are still some disagreements. Many researchers use the “H₂ production per unit hour ($\text{mmol}\cdot\text{h}^{-1}$)” to express the photocatalytic activity (*Nano Energy* 85 (2021) 105949; *Applied Catalysis B: Environmental* 267 (2020) 118651; *Applied Catalysis B: Environmental* 270 (2020) 118855; *Nature Chemistry*, 12, 1150-1156(2020); *Adv. Funct. Mater.* 2020, 2003007; *Nano Energy* 76 (2020) 104972; *Applied Catalysis B: Environmental* 278 (2020) 119253; *Adv. Funct. Mater.* 2020, 2003731). Meanwhile, there are also a large number of researchers adopt the representation of “H₂ production per unit weight of photocatalyst per unit time ($\text{mmol}\cdot\text{g}^{-1}\cdot\text{h}^{-1}$)” to express the photocatalytic activity (*Nature Materials*. 19, 559-565(2020); *Nature Communications*, 12, 483 (2021); *J. Am. Chem. Soc.* 2020, 142, 9752-9762; *Applied Catalysis B: Environmental* 291 (2021) 120139; *ACS Catal.* 2021, 11, 7, 4362-4371; *Angew. Chem. Int. Ed.* 2021, 60, 15, 8236-8242; *Adv. Mater.* 2021, 33, 2003327; *Adv. Funct. Mater.* 2021, 2100816;

Adv. Mater. 2020, 32, 1906361; *Nano Energy* 81 (2021) 105608.). As known from the above literatures, there is no unified format for the expression of photocatalytic rate presently, and we can't say which one is wrong.

In our limited knowledge, the above two rate expressions are suitable for the two development stages of photocatalytic water splitting reaction, respectively, that is, the laboratorial stage (the stage we are in currently) and the industrialization stage (in the future). As for the laboratorial stage, the major challenge we faced now, is the lack of cheap, efficient and stable photocatalyst. Therefore, the screening of efficient photocatalytic materials by investigating their activity data per weight (even per active sites) is necessary. Under this circumstance, the low concentration of photocatalyst usage which can provide high photocatalyst or active sites utilization, is more favorable to reflect the real activity of photocatalyst. There are a large number of literatures adopting the low concentration of photocatalyst usage, such as, *Nature Communications*, 12, 483 (2021) (0.2 g/L); *Nature Materials* 19, 559-565 (2020) (0.1 g/L); *Nano Today* 37 (2021) 101080 (0.2 g/L); *Appl. Catal. B Environ.* 289, 120040 (2021) (0.5 g/L); *Adv. Mater.* 2021, 33, 2003327 (0.1 g/L); *Appl. Catal. B Environ.* 291, 120104 (2021) (0.125 g/L); *ACS Catal.* 11, 4, 2098-2107 (2021) (0.1 g/L); *Adv. Mater.* 2020, 2004561 (0.4 g/L); *Adv. Funct. Mater.* 2020, 1910830 (0.25 g/L); *Nano Energy* 76 (2020) 105031 (0.2 g/L). Therefore, in our work, for case of making comparison between our photocatalytic performance with that of other literatures, the relatively low photocatalyst concentration was adopted (0.5 g/L). In addition, due to that all the photocatalytic activity in our work were carried out under the same conditions (photocatalyst concentration, temperature, light source, sacrificial agent and et al.), thus the photocatalytic results between different photocatalysts have a certain comparative significance.

However, in the future, when the ideal catalyst (cheap, efficient and stable) is developed, the photocatalytic activity expression of “H₂ production per unit time (mmol·h⁻¹)” maybe more suitable than that of “H₂ production per unit weight of photocatalyst per unit time (mmol·g⁻¹·h⁻¹)”. Because by then, the research focus would be transferred from “photocatalyst itself” to “efficient solar energy utilization”. In this stage, increasing the catalyst usage to absorb as much sunlight as possible to carry out efficient conversion and production of sustainable energy (solar energy and H₂ energy) would be of great significance for the practical industrial application of photocatalytic technology. Thanks again for the reviewer providing us this

enlightening comment.

Comment 1.9

There are many unclear statements and mistakes/errors, e.g., “are becoming increasing”, “momentous significance”, “Z-scheme” vs. “Z-Scheme”, “continuously modulate of Z-scheme”.

Response:

We express our sincere appreciation for the suggestion. According to the reviewer’s suggestion, we have check out the manuscript carefully, and solved all the problems the reviewer mentioned point to point. In addition, we have consulted a professional to polish the writing. Thanks again for the reviewer’s kind comment.

Comment 1.10

Additionally, I think that “Vs” term is confusing, as sulfur symbol should be written as “S” (from capital letter). I do recommend changing this into “vS”, “Sv” or “Svac.”

Response:

Thanks for this careful and nice comment. We have considered the reviewer’s suggestion, and decided to transform the abbreviation of S-vacancies to “S_v” in the whole manuscript. We express our sincere appreciation for the valuable suggestion.

Comment 1.11

Moreover, I wonder if similar research has been performed only in China – only 5 references out of ca. 50 outside China.

Response:

Thanks for the reviewer’s prudential review. In our article, the references cited in our manuscript are usually the most cutting-edge and the most relevant from the high-impact journals. During quoting, we didn't throw attention on the author and nationality. Additionally, in the course of completing this revision, some related articles are added to the revised manuscript, which has also been listed as follows.

3. Yang, J. L. et al. Boosting Photocatalytic Hydrogen Evolution Reaction Using Dual Plasmonic Antennas. *ACS Catal.*, **11**, 5047-5053 (2021).

7. Kageshima, Y, et al, Photocatalytic and photoelectrochemical hydrogen evolution from water over $\text{Cu}_2\text{Sn}_x\text{Ge}_{1-x}\text{S}_3$ particles. *J. Am. Chem. Soc.* (2021). <https://doi.org/10.1021/jacs.0c12140>.
8. Wang, S. B. et al. Supporting Ultrathin ZnIn_2S_4 Nanosheets on Co/N-Doped Graphitic Carbon Nanocages for Efficient Photocatalytic H_2 Generation, *Adv. Mater.* **31**, 1903404 (2019).
9. Kim, Y. et al. Efficient photocatalytic production of hydrogen by exploiting the polydopamine-semiconductor interface, *Appl. Catal. B: Environ.* **280**, 119423 (2021).
10. Li, Z. J. et al. CoNi Bimetal Cocatalyst Modifying a Hierarchical ZnIn_2S_4 Nanosheet-Based Microsphere Noble-Metal-Free Photocatalyst for Efficient Visible-Light-Driven Photocatalytic Hydrogen Production. *ACS Sustain. Chem. Eng.* **7**, 20190-20201 (2019).
11. Mohanty, B. Chattopadhyay, A. Nayak, J. Band gap engineering and enhancement of electrical conductivity in hydrothermally synthesized CeO_2 -PbS nanocomposites for solar cell applications. *J. Alloys Compd.* **850**, 156735 (2021).
25. Xing, F. S. et al. Tunable charge transfer efficiency in $\text{H}_x\text{MoO}_3@\text{ZnIn}_2\text{S}_4$ hierarchical direct Z-scheme heterojunction toward efficient visible-light-driven hydrogen evolution. *Appl. Catal. B Environ.* **285** 119818 (2021).

Response to Reviewer #2

Comment 2.1

Sulphur vacancies were created by introduction of $\text{N}_2\text{H}_4\cdot\text{H}_2\text{O}$ during ZnIn_2S_4 synthesis, however, only one stoichiometric ration (between $\text{N}_2\text{H}_4\cdot\text{H}_2\text{O}$ and other reagents) have been used. Why? How did the authors choose the amount of hydrazine introduced into reaction environment?

Response:

We express our sincere appreciation for this valuable suggestion. In our manuscript, $\text{N}_2\text{H}_4\cdot\text{H}_2\text{O}$ was added to induce the generation of coordination unsaturated S atoms and sulfur (S) vacancies. In fact, we had investigated the addition amount of $\text{N}_2\text{H}_4\cdot\text{H}_2\text{O}$ on the photocatalytic activity of $\text{S}_v\text{-ZnIn}_2\text{S}_4/\text{MoSe}_2$, and tested the H_2 evolution efficiency (the results are showing in **Fig. R2**). As observed, when the addition of $\text{N}_2\text{H}_4\cdot\text{H}_2\text{O}$ was 1.5 mL, the H_2 production rate of $\text{S}_v\text{-ZnIn}_2\text{S}_4/\text{MoSe}_2$ presents the highest, demonstrating that the 1.5 mL $\text{N}_2\text{H}_4\cdot\text{H}_2\text{O}$ could lead to the most suitable coordination unsaturated S atoms concentration and S-vacancies in $\text{S}_v\text{-ZnIn}_2\text{S}_4/\text{MoSe}_2$, which can influence the photocarriers separation and interfacial contact state between $\text{S}_v\text{-ZnIn}_2\text{S}_4$ and MoSe_2 . However, in our work, the research

emphasis is to modulate the Z-scheme charge transfer by constructing an atomic-level interfacial interface (Mo-S) and internal electric field, and realize efficient photocatalytic H₂ evolution performance. Therefore, in order to highlight the key points and control the length of the manuscript, we haven't added the discussion about changing N₂H₄·H₂O addition, just put the optimized results into the manuscript. Thanks again for this valuable suggestion.

Fig. R2. H₂ evolution rate of S_v-ZIS/5.0MoSe₂ synthesized by adding different amount of N₂H₄·H₂O.

Comment 2.2

Photolysis results of sacrificial agent in hydrogen generation should be given

Response:

Thanks for the reviewer's professional comment. According to the reviewer's suggestion, we have tested the H₂ evolution rate of sacrificial agent solution (ascorbic acid, AA) in the absence of any photocatalyst. As observed in **Fig. R3**, after four hours of light, no peak corresponding to H₂ can be detected by the gas chromatograph (the H₂ peak should be at around 1.445 min), illustrating that the AA cannot split water to generate H₂. Thanks again for this nice suggestion.

Fig. R3. The image of chromatography workstation for the photocatalytic system of ascorbic acid solution.

Comment 2.3

In figure 6d, 1%C@HPW/CdS; 2%C@HPW/CdS; 3%C@HPW/CdS; 5%C@HPW/CdS and 7%C@HPW/CdS samples are listed (containing different amount of HPW), but these sample are not mentioned in any other part of the manuscript.

Response:

Thanks for the reviewer's comment. However, these samples mentioned by the reviewer are not the materials used in our work. The material system in our manuscript is $S_v\text{-ZnIn}_2\text{S}_4/\text{MoSe}_2$.

Comment 2.4

How the amount of sulphur vacancies could be controlled?

Response:

We express our sincere appreciation for this valuable suggestion. In our work, the generation of sulfur (S) vacancies depends on the addition of $\text{N}_2\text{H}_4\cdot\text{H}_2\text{O}$, which can

release a mass of reductive gas including N_2 , H_2 and NH_3 during hydrothermal reaction, and these gasses would destruct the covalent bond of Zn-S, and generating a large amount of coordination unsaturated S atoms, when the covalent bond was broken furtherly, S-vacancies would be generated. Therefore, the amount of S-vacancies can be adjusted by changing the addition of $N_2H_4 \cdot H_2O$. During our experiments, the S_v - $ZnIn_2S_4/MoSe_2$ photocatalyst synthesized by adding different $N_2H_4 \cdot H_2O$ amount was synthesized, and tested the H_2 evolution efficiency (the results are showing in **Fig. R4**). As observed, when the addition of $N_2H_4 \cdot H_2O$ was 1.5 mL, the H_2 production rate of S_v - $ZnIn_2S_4/MoSe_2$ presents the highest, demonstrating that the 1.5 mL $N_2H_4 \cdot H_2O$ could lead to the most suitable coordination unsaturated S atoms concentration and S-vacancies in S_v - $ZnIn_2S_4/MoSe_2$, which can influence the photocarriers separation and interfacial contact state between S_v - $ZnIn_2S_4$ and $MoSe_2$. However, the primary target in our work is to construct an interfacial chemical bond and internal electric field comodulated Z-scheme heterostructure photocatalyst for efficient photocatalytic H_2 evolution, rather than introduce S-vacancies in $ZnIn_2S_4$. Therefore, in order to highlight the key points and control the length of the manuscript, we didn't add the discussion about adjusting the amount of $N_2H_4 \cdot H_2O$ into our manuscript. Thanks again for this valuable suggestion.

Fig. R4. H_2 evolution rate of S_v -ZIS/5.0 $MoSe_2$ synthesized by adding different amount of $N_2H_4 \cdot H_2O$.

Response to Reviewer #3

Comment 3.0

The authors have synthesized a sulfur vacancy rich $ZnIn_2S_4$ and $MoSe_2$

heterostructure through in-situ growth for efficient photocatalytic hydrogen evolution. They have demonstrated that the intimate interfacial connection and Z-scheme charge transfer system between MoSe₂ and ZnIn₂S₄ allows for a markedly improved hydrogen evolution rate. The manuscript contains thorough discussions, along with well-rounded characterization and performance analysis of the synthesized photocatalyst. However, since sulfur vacancies in ZnIn₂S₄ (Appl. Cat. B 2019, 248, 193-201; Phys. Chem. Chem. Phys. 2019, 21, 25484-25494; J. Energy Chem 2021, 58, 397-407) and the formation of its heterojunction with MoSe₂ (CrystEngComm 2021, 10.1039/D0CE01808B; ChemSusChem 2017, 10, 4624-4631) in photocatalytic hydrogen evolution have been widely reported in recent years, I do not recommend this manuscript to be published in high-impact Nature Communications owing to limited novelty, urgency and scientific merits. The hybrid system is not something new and the synthesis strategy is rather not novel (as claimed in comment #1). Despite extensive characterization analysis and superb photocatalytic performance, this work does not provide new scientific knowledge which can merit wide readership in this field. Nevertheless, I believe this manuscript would be suitable for other publications such as Communications Chemistry. To further improve the quality of the paper, some of the comments are listed below.

Response

We sincerely appreciate the reviewer's insightful comments, which greatly help us better organize and improve the quality of our manuscript. However, we think that the overall innovation in our manuscript was not be felicitously evaluated, because there is fundamental difference between our work and these literatures mentioned by the reviewer.

In detail, in the mentioned articles, Du et al. synthesized a monolayer ZnIn₂S₄ with sulfur vacancies through an ethanol/water mixed solvothermal method (*Applied Catalysis B: Environmental* 248 (2019) 193-201). Wang et al. prepared a hydrogenated ZnIn₂S₄ (H-ZIS) microspheres with surface-deficient porous structures through a hydrothermal followed by hydrogenation (at 300°C) strategy (*Phys. Chem. Chem. Phys.* 2019, 21, 25484-25494). Jing et al. developed an S-defect-controlled

ZnIn₂S₄ photocatalyst via adjusting the content of L-Cysteine during hydrothermal process (*Journal of Energy Chemistry* 58 (2021) 397-407). Although both our work and the work in the literatures mentioned above involve the introduction of S vacancies in ZnIn₂S₄, the focus of attention is absolutely different. In the articles mentioned by the reviewer, authors synthesized the S-vacancies abundant ZnIn₂S₄ photocatalyst, where the S-vacancies can serve as trapping centers for photogenerated electrons, thus facilitating the photocarriers separation, and then improving the photocatalytic performance of ZnIn₂S₄. In our work, although S-vacancies were also generated in ZnIn₂S₄, which played the role of electron “traps” for accelerating the photocarriers separation process, the primary purpose of our work is to create abundant coordination unsaturated S atoms on the surface of ZnIn₂S₄ to serve as the anchoring sites for the nucleation and growth of MoSe₂, thereby constructing a chemical bond (Mo-S) connected heterostructure interface, rather than to merely produce S-vacancies in ZnIn₂S₄. The interfacial Mo-S bond can serve as efficient charge shuttle channel with small charge transfer resistance, thus benefiting for the formation of intense internal electric field, which can provide the necessary driving force for the Z-scheme charge transfer inside ZnIn₂S₄/MoSe₂ heterostructure. In other words, the S-vacancies in our material system can only be viewed as the result of the further decrease in the coordination number of unsaturated S atoms. Therefore, from this point of view, our work is completely different with the above three articles. The innovation of our work mainly lies on constructing an efficient Z-scheme photocatalyst through creating an intimate atomic-level interfacial combination between two different semiconductors with suitable electronic structure. Moreover, the generation method of S-vacancies between our work with the reviewer mentioned articles is also different. In our work, the generation of S-vacancies in ZnIn₂S₄ depends on the addition of strongly reductive hydrazine monohydrate (N₂H₄·H₂O), which could release reductive gases (such as, N₂, H₂) during hydrothermal reaction. The released reductive gases could attack the covalent bond (mainly Zn-S) in ZnIn₂S₄, and give rise to the bond breaking. In this way, S atom in ZnIn₂S₄ would possess abundant unsaturated bond, and when Zn-S bond was further break, the S-vacancies

would be formed.

In addition, it is true that the materials system in our work is as same as Feng's and Zeng's group (*CrystEngComm* 2021, 10.1039/D0CE01808B; *ChemSusChem* 2017, 10, 4624-4631), however, due to the distinct-different technology conditions, there are huge distinctions in structure, mechanism and property between our work and the two works illustrated by the reviewer. The reasons are stated as follows.

Firstly, from the perspective of microstructure, our work is superior to Feng's (*CrystEngComm* 2021, 10.1039/D0CE01808B) and Zeng's work (*ChemSusChem* 2017, 10, 4624-4631). In detail, in our work, $\text{N}_2\text{H}_4\cdot\text{H}_2\text{O}$ was added during hydrothermal process, which can release reductive gases (such as, N_2 and H_2) under high temperature. These released gasses can break the covalent bond (mainly Zn-S), leading to the generation of abundant coordination unsaturated S atoms in ZnIn_2S_4 . When Zn-S bond was further break, these S atoms would escape from ZnIn_2S_4 and generate S-vacancies. The S-vacancies can serve as electron traps promoting the separation of photocarriers in ZnIn_2S_4 , therefore contributing to photocatalytic H_2 evolution. Most importantly, in our work, the ample coordination unsaturated S atoms can provide excellent anchoring sites for the attachment of MoSe_2 on the surface of ZnIn_2S_4 through Mo-S bond (which has been fully verified by XPS and Raman results), thus forming the atomic-level interface contact between ZnIn_2S_4 and MoSe_2 . However, in Feng's work, the coordination unsaturated S atoms and S-vacancies were not formed in ZnIn_2S_4 , therefore, the combination between ZnIn_2S_4 and MoSe_2 can only be the simple physical connect, rather than the atomic-level interface contact. Besides, from the morphology of the $\text{ZnIn}_2\text{S}_4/\text{MoSe}_2$, as observed from the SEM image in Fig.4(a) of Feng's article, the size of $\text{MoSe}_2/\text{ZnIn}_2\text{S}_4$ microsphere is about 6.8 μm , and the accumulation of ZnIn_2S_4 and MoSe_2 is very dense. Besides, it can be noted that the ZnIn_2S_4 and MoSe_2 are existing individually (as observed in the TEM image in Fig. 4(c) of Feng's article). In comparison, as observed in Fig. 2 in our manuscript, the size of $\text{S}_v\text{-ZnIn}_2\text{S}_4/\text{MoSe}_2$ microspheres are about 1 μm , and the nanosheets that make up $\text{S}_v\text{-ZnIn}_2\text{S}_4/\text{MoSe}_2$ flower-like microspheres are very loose and thin. Additionally, we cannot distinguish the heterojunction interface of ZnIn_2S_4 and MoSe_2 from the TEM image of $\text{ZnIn}_2\text{S}_4/\text{MoSe}_2$, and only the lattice fringe in HRTEM can prove that MoSe_2 was attached on the surface of ZnIn_2S_4 intimately. What's more, in Zeng's work, the ZnIn_2S_4 and MoSe_2 were synthesized separately,

then mixed by a solution-phase hybridization method at room temperature. It is obvious that the simple physical mixture cannot create an atomic-level interfacial combination, thus leading to the independent existence of ZnIn_2S_4 and MoSe_2 (as observed from the TEM image of Fig. 3(c) in Zeng's article). Under this circumstance, not only the interfacial charge transfer impedance inside $\text{ZnIn}_2\text{S}_4/\text{MoSe}_2$ would be great, but also the overall structure stability of $\text{ZnIn}_2\text{S}_4/\text{MoSe}_2$ cannot be well maintained. As known from the above analysis, benefiting from the specific preparation process, the $\text{ZnIn}_2\text{S}_4/\text{MoSe}_2$ heterostructure in our work presents novel interfacial combination mode and microstructure, which is one of the major innovations of our work, and must result in the specific photocatalytic mechanism and property.

Secondly, from the perspective of photocatalytic mechanism, our work is more advanced. Specifically, in Feng's article, under light irradiation, the electrons in the conduction band (CB) of ZnIn_2S_4 with high potential transfer to the CB of MoSe_2 with low potential, while the holes in the valence band (VB) of MoSe_2 transfer to the VB of ZnIn_2S_4 . Although the photocarriers were separated in different component, it should be noted that the reducing ability of photogenerated electrons were greatly weakened, which is probably one of the reasons for the poor photocatalytic performance of Feng's photocatalyst. Meanwhile, in Zeng's article, due to the simple physical combination and large charge transfer resistance, the MoSe_2 can only serve as cocatalyst for drawing photogenerated electrons in ZnIn_2S_4 , which exhibits limited photocarriers separation efficiency. In contrast, in our work, owing to the intimate interfacial chemical bonding effect and the different energy band between $\text{S}_v\text{-ZnIn}_2\text{S}_4$ and MoSe_2 , the strong internal electric field would be fabricated at the contact interface of $\text{S}_v\text{-ZnIn}_2\text{S}_4$ and MoSe_2 . Under the action of internal electric field, the photogenerated electrons in the CB of MoSe_2 would transfer to the VB of $\text{S}_v\text{-ZnIn}_2\text{S}_4$ and recombine with the photogenerated holes, that's the Z-scheme charge transfer mechanism, which has been fully verified by the surface photovoltage and DMPO spin-trapping EPR experiment. Under this mechanism, not only the separation and transfer of photocarriers can be signally promoted, but also the photogenerated electrons with high reducing ability in the CB of $\text{S}_v\text{-ZnIn}_2\text{S}_4$ would be reserved, thus

contributing to the outstanding photocatalytic H₂ evolution performance of S_v-ZnIn₂S₄/MoSe₂. In addition, S-vacancies in ZnIn₂S₄ can serve as electron traps, further promoting the H₂ evolution reaction. In conclusion, the excellent photocatalytic activity of S_v-ZnIn₂S₄/MoSe₂ in our work was mainly attributed to the interfacial chemical bond and internal electric field comodulated Z-scheme charge transfer mechanism. Additionally, S-vacancies in S_v-ZnIn₂S₄/MoSe₂ also played a certain role in promoting photocatalytic activity. The above conclusions have been revealed in-depth and verified in our manuscript by various experimental and characterization results.

Thirdly, due to the different photocatalytic mechanism, the H₂ evolution property of our work and that of Feng's and Zeng's are different, and our work is obviously more prominent. In Feng's article, the photocatalytic H₂ evolution rate and apparent quantum yield (AQY) of the optimized ZnIn₂S₄/MoSe₂ photocatalyst are 1226 $\mu\text{mol}\cdot\text{h}^{-1}\cdot\text{g}^{-1}$ and 42.3%, respectively. As for Zeng's article, the optimized ZnIn₂S₄/MoSe₂ performs a H₂ generation rate of 2228 $\mu\text{mol}\cdot\text{h}^{-1}\cdot\text{g}^{-1}$ and an AQY of 21.39% at 420 nm. In comparison, the H₂ evolution rate of the optimized S_v-ZnIn₂S₄/MoSe₂ photocatalyst in our work reaches to 63210 $\mu\text{mol}\cdot\text{h}^{-1}\cdot\text{g}^{-1}$, which is about 51.6 and 28.4 times higher than that of Feng's and Zeng's work. What's more, the AQY of our S_v-ZnIn₂S₄/MoSe₂ photocatalyst is 76.48% at 420 nm, also significantly higher than that of Feng's and Zeng's photocatalyst.

Given the above discuss, we think that the literatures cited by the reviewer cannot weaken the novelty of our work. We have present novel and profound comprehend on experimental principle, interfacial structure, photocatalytic mechanism and property, which would evoke a lot new realization on developing novel and efficient photocatalytic material with interfacial chemical bond and internal electric field co-modulated Z-scheme charge transfer mechanism. Thanks again for the reviewer's comment.

Comment 3.1

In another study done by Feng et al., (CrystEngComm 2021, Constructing 2D/2D

Heterojunction of MoSe₂/ZnIn₂S₄ Nanosheets for Enhanced Photocatalytic Hydrogen Evolution, 10.1039/D0CE01808B), their proposed heterojunction system was that of a p-n system despite using a similar in-situ formation strategy of MoSe₂, albeit without the added modification of the S vacancy. Could the authors provide some insights as to why this could be? It is important to have clear and confirmed charge transfer mechanism.

Response:

We express our sincere appreciation for this meticulous comment on our manuscript. It is true that Feng's group adopted the similar strategy as ours to fabricate ZnIn₂S₄/MoSe₂ heterostructure, however, due to the different technology process conditions, there are huge distinction in structure, mechanism and property between our work and Feng's.

To be specific, in Feng's work, only the simple physical contact was constructed between ZnIn₂S₄ and MoSe₂, rather than form the chemical bond connection. This weak physical connection would give rise to the great charge transfer resistance between ZnIn₂S₄ and MoSe₂, thus the electrons in ZnIn₂S₄ can hardly transfer to MoSe₂, and the internal electric field cannot be formed in MoSe₂/ZnIn₂S₄ heterojunction. In this circumstance, under light irradiation, the photogenerated electrons in the conduction band (CB) of ZnIn₂S₄ with high potential can only transfer to the CB of MoSe₂ with low potential, leading to the significantly decrease of the reducing ability of photogenerated electrons, which is not conducive to the photocatalytic H₂ evolution. Therefore, the MoSe₂/ZnIn₂S₄ photocatalyst in Feng's work presents a very low H₂ production rate of 1226 μmol·h⁻¹·g⁻¹ and a small apparent quantum yield (AQE) of 42.3%.

In contrast, under the suitable technological conditions, the N₂H₄·H₂O added in our system can release abundant reductive gases (such as, N₂ and H₂), which can attack the covalent bond of Zn-S, leading to the generation of abundant coordination unsaturated S atoms. When Zn-S bond was further break, these unsaturated S atoms would escape from ZnIn₂S₄ and generate S-vacancies. The S-vacancies can serve as electron traps promoting the photocarriers separation in ZnIn₂S₄, and therefore contributing to photocatalytic H₂ evolution rate. More importantly, the ample coordination unsaturated S atoms can provide the excellent anchoring sites for the attachment of MoSe₂ on the surface of ZnIn₂S₄ through Mo-S bond, thus forming the atomic-level interface contact interface between ZnIn₂S₄ and MoSe₂. In this case, the

free electrons in ZnIn_2S_4 can swimmingly transfer to MoSe_2 along the Mo-S bond, leading to that the interface near ZnIn_2S_4 side possess positive charge, while negative charge existed in the side of MoSe_2 , as a result, an internal electric field from ZnIn_2S_4 to MoSe_2 would be built. Under the irradiation of visible light, the electrons in the CB of MoSe_2 would migrate to the VB of ZnIn_2S_4 to recombine with the holes under the driving effect of the internal electric field, that's the Z-scheme charge transfer mechanism, which has been fully verified by the surface photovoltage and DMPO spin-trapping EPR experiment. Under this mechanism, not only the separation and transfer of photocarriers can be signally promoted, but also the photogenerated electrons with high reducing ability in the CB of ZnIn_2S_4 be reserved. Benefiting from the specific interfacial structure and Z-scheme charge transfer mechanism, the $\text{S}_v\text{-ZnIn}_2\text{S}_4/\text{MoSe}_2$ photocatalyst in our work performs the outstanding H_2 evolution rate of $63210 \mu\text{mol}\cdot\text{h}^{-1}\cdot\text{g}^{-1}$ (about 51.6 times higher than that of Feng's work) and a high AQY of 76.48% at 420 nm, also prominently higher than that of Feng's photocatalyst.

Given the above discuss, due to the different technology conditions and interfacial combination mode, the photocatalytic mechanism presents distinct difference between Feng's work and ours. Furtherly, under the different photocatalytic mechanism, the $\text{S}_v\text{-ZnIn}_2\text{S}_4/\text{MoSe}_2$ photocatalyst in our work presents predominant superiority in H_2 evolution activity. Thanks again for the reviewer's careful and professional comment.

Comment 3.2

The authors have shown that although the light absorption is highest with a mass ratio of MoSe_2 to ZIS is 7%, the H_2 production is not the maximum as it has less effective photocarrier separation as compared to the 5% mass ratio. Could the authors comment on the underlying mechanism behind this? Evidence should be supplemented to strengthen the statement.

Response:

Many thanks for the reviewer's careful comment. In our work, the highest light absorption of $\text{S}_v\text{-ZnIn}_2\text{S}_4/\text{MoSe}_2$ with the mass ratio of MoSe_2 to ZnIn_2S_4 of 7% was caused by the highest amount of MoSe_2 (with black color) in $\text{S}_v\text{-ZnIn}_2\text{S}_4/7.0\text{MoSe}_2$. However, in addition to the light absorption, the separation and transfer efficiency

also play important influence on photocatalytic performance. In order to reveal the reason for the optimal photocatalytic performance of $S_v\text{-ZnIn}_2\text{S}_4/\text{MoSe}_2$ with the mass ratio of MoSe_2 to $S_v\text{-ZnIn}_2\text{S}_4$ of 5%, besides to UV-vis absorbance spectra, the photoluminescence spectra and photocurrent response of $S_v\text{-ZnIn}_2\text{S}_4/\text{MoSe}_2$ with different mass ratio of MoSe_2 to $S_v\text{-ZnIn}_2\text{S}_4$ ($S_v\text{-ZIS}/0.5\text{MoSe}_2$, $S_v\text{-ZIS}/1.0\text{MoSe}_2$, $S_v\text{-ZIS}/3.0\text{MoSe}_2$, $S_v\text{-ZIS}/5.0\text{MoSe}_2$, $S_v\text{-ZIS}/7.0\text{MoSe}_2$) were further tested, the results are showing in **Fig. R5**. As observed in **Fig. R5(a)**, under 375 nm excitation wavelength, the $S_v\text{-ZIS}/5.0\text{MoSe}_2$ exhibits the lowest PL emission peak intensity, illustrating the most efficient photocarriers separation efficiency of $S_v\text{-ZIS}/5.0\text{MoSe}_2$. **Fig. R5(b)** is photocurrent response of different photocatalyst, it can be noted that under the FX-300 Xe lamp, $S_v\text{-ZIS}/5.0\text{MoSe}_2$ presents the highest photocurrent density among all the tested photocatalysts, suggesting the most efficient photocarriers transfer ability of $S_v\text{-ZIS}/5.0\text{MoSe}_2$. As known from the above results, we can conclude that even the light absorption of $S_v\text{-ZIS}/5.0\text{MoSe}_2$ is not the highest, the separation and transfer efficiency of $S_v\text{-ZIS}/5.0\text{MoSe}_2$ is the most outstanding, which contribute to the efficient photocatalytic H_2 evolution rate. Thanks again for this professional comment.

Fig. R5. (a) Photoluminescence spectra (PL, excited at 375 nm) and (b) photocurrent response of $S_v\text{-ZIS}/\text{MoSe}_2$ with different mass ratio of MoSe_2 to ZIS.

Fig. R5 were shown in the Supporting Information of Fig. S8 and S9, the following description was also added into the revised manuscript on Line 18-23/Page 11/Page 18 to Line 1-12/Page 19 and highlighted:

In order to investigate the effects of the MoSe_2 to ZIS mass ratio on the

photocatalytic performance of S_v -ZIS/MoSe₂ composites. The light absorption, photocarriers separation and photocurrent density of S_v -ZIS/MoSe₂ photocatalysts with different mass ratio of MoSe₂ to ZIS were also characterized by UV-vis absorption, steady-state PL spectroscopy and photocurrent response. As observed in Fig. S7, with increasing the mass ratio of MoSe₂ to ZIS, the light absorption intensity enhance gradually. It is worth mentioning that the S_v -ZIS/7.0MoSe₂ sample displays the strongest light absorption ability, but its photocatalytic H₂ production performance is not the best (as known from Fig. 4a), suggesting that the light absorption is not the only decisive factor for the photocatalytic activity. Fig. S8 is the PL spectra, it can be observed that the PL peak of S_v -ZIS/5.0MoSe₂ is the lowermost, revealing the most effective photocarriers separation when the mass ratio of MoSe₂ to ZIS is 5%, which directly explains why the S_v -ZIS/5.0MoSe₂ sample has the best photocatalytic performance. Fig. S9 shows the photocurrent response. As displayed, the S_v -ZIS/5.0MoSe₂ shows the highest photocurrent density, which is the result of high-efficiency separation and transfer of photogenerated electron and hole, further revealing the optimum photocatalytic performance of S_v -ZIS/5.0MoSe₂. As known from the above results, the prominent photocatalytic performance requires the coordination among the efficient light absorption, photocarrier separation and transfer ability.

Comment 3.3

From Figure 6a, the band gap of ZIS, V_s -ZIS and V_s -ZIS/MoSe₂ was 2.35, 2.28 and 2.19 eV, respectively. Why do the authors state the decrease in band gap 2.28 to 2.19 eV after the introduction of MoSe₂ when the heterojunction system consists of the individual band gaps of the two components (V_s -ZIS and MoSe₂) as shown in Figure 7a? How do the authors clarify the 2.19 eV band gap for the V_s -ZIS/MoSe₂.

Response:

Many thanks for the reviewer's insightful comment. In our manuscript, the band gap (E_g) value was determined by the Kubelka-Munk function vs. the energy of incident light plots, which was transformed by the tested UV-vis absorption spectrum, and the E_g of ZIS, S_v -ZIS and S_v -ZIS/MoSe₂ were determined to be 2.35, 2.28 and 2.19 eV,

respectively. In addition, based on the UPS spectra (Fig. 6e in the revised manuscript), the work function (ϕ) is of S_v-ZIS and MoSe₂ can be determined to be 17.65 and 16.87 eV, respectively. Accordingly, the fermi level (E_f) of S_v-ZIS and MoSe₂ can be determined as -0.93 and -0.15 V, respectively. The lower E_f of MoSe₂ than that of S_v-ZIS means that the free electrons in S_v-ZIS would transfer to MoSe₂ when they contact intimately. The electron drifting from S_v-ZIS to MoSe₂ results in the charge redistribution on the interface between S_v-ZIS and MoSe₂, in which the interface near S_v-ZIS side possess positive charge, while negative charge existed in the side of MoSe₂. As a result, a new equilibrium state E_f would be fabricated between S_v-ZIS and MoSe₂. Meanwhile, along with the electrons drifting, the band structure of S_v-ZIS bends upward, while MoSe₂ bends downward, thus contributing to a new and narrower band gap of S_v-ZIS/MoSe₂ than that of S_v-ZIS but broader than that of MoSe₂. Moreover, in order to clearly reflect the band structure changes and the formation of internal electric field, as well as to investigate the charge transfer pathway and photocatalytic mechanism of S_v-ZIS/MoSe₂, the respective band structure S_v-ZIS and MoSe₂ was depicted in Fig. 7a. According to the reviewer's suggestion, we have added the band structure of S_v-ZIS/MoSe₂ with E_g of 2.19 eV in the mechanism illustration in Fig. 7a, which has also been displayed in the following **Fig. R6**. Thanks again for the reviewer's valuable suggestion.

Fig. R6. Photocatalytic reaction mechanism of S_v-ZIS/MoSe₂ under light irradiation.

Comment 3.4

It is noted that with the presence of S vacancy, the absorption edge is extended to

higher wavelength of light. Could the authors include the action spectra analysis in the performance measure of the synthesized catalyst?

Response:

Thanks for the reviewer's valuable suggestion. Action spectra analysis is a plot of apparent quantum yield (AQY) against incident light wavelength, which can be used for reflecting the light absorption and conversion ability of efficient photocatalyst. Therefore, in the original manuscript, the H₂ evolution per unit time of S_v-ZIS/MoSe₂ under different monochromatic light ($\lambda=380, 420, 500$ and 600 nm) was tested, and the light power of 300 W Xe lamp with different bandpass filter were measured by PLMW2000 photoradiometer. The test results and the detailed calculation process of AQY are displaying as follows, which has also been presented in the Supporting Information of our manuscript.

Table R1. The H₂ evolution per time of S_v-ZIS/MoSe₂ photocatalyst and light power at different monochromatic wavelengths

Wavelength	The mean H ₂ production in 1 hour	Light power (Xe lamp)
380 nm	90.46 μmol	0.034 W
420 nm	258.53 μmol	0.107 W
500 nm	156.66 μmol	0.140 W
600 nm	0.88 μmol	0.134 W

Table R1 was shown in the Supporting Information of our manuscript as Table S5. And the AQY of S_v-ZIS was calculated based on the following equation:

$$\begin{aligned}
 AQY &= \frac{\text{the number of reacted electrons}}{\text{the number of incident photons}} \times 100\% \\
 &= \frac{4 \times \text{the number of H}_2 \text{ molecules}}{\text{the number of incident photons}} \times 100\% \\
 &= \frac{4 \times \text{the number of H}_2 \text{ molecules}}{N} \times 100\%
 \end{aligned}$$

For different monochromatic light wavelength, the detailed calculating process are presenting as follows:

$\lambda=380$ nm:

$$N = \frac{E\lambda}{hc} = \frac{0.034 \times 3600 \times 380 \times 10^{-9}}{6.626 \times 10^{-34} \times 3 \times 10^8} = 2.34 \times 10^{20}$$

$$AQY = \frac{4 \times 90.46 \times 10^{-6} \times 6.02 \times 10^{23}}{2.34 \times 10^{20}} \times 100\% = 93.08\%$$

$\lambda=420$ nm:

$$N = \frac{E\lambda}{hc} = \frac{0.107 \times 3600 \times 420 \times 10^{-9}}{6.626 \times 10^{-34} \times 3 \times 10^8} = 8.14 \times 10^{20}$$

$$AQY = \frac{4 \times 258.53 \times 10^{-6} \times 6.02 \times 10^{23}}{8.14 \times 10^{20}} \times 100\% = 76.48\%$$

$\lambda=500$ nm:

$$N = \frac{E\lambda}{hc} = \frac{0.140 \times 3600 \times 500 \times 10^{-9}}{6.626 \times 10^{-34} \times 3 \times 10^8} = 1.46 \times 10^{21}$$

$$AQY = \frac{4 \times 156.66 \times 10^{-6} \times 6.02 \times 10^{23}}{1.27 \times 10^{21}} \times 100\% = 29.70\%$$

$\lambda=600$ nm:

$$N = \frac{E\lambda}{hc} = \frac{0.134 \times 3600 \times 600 \times 10^{-9}}{6.626 \times 10^{-34} \times 3 \times 10^8} = 1.46 \times 10^{21}$$

$$AQY = \frac{4 \times 0.88 \times 10^{-6} \times 6.02 \times 10^{23}}{1.46 \times 10^{21}} \times 100\% = 0.15\%$$

Based on the above calculation, the wavelength-dependent AQY plot (action spectrum) was depicted and displayed in **Fig. R7**. As observed, the action spectrum of S_v-ZIS/MoSe₂ matches well with the UV-vis absorption spectra, indicating the outstanding optical absorption and utilization capacity of S_v-ZIS/MoSe₂ photocatalyst.

Fig. R7. Wavelength-dependent apparent quantum yield (AQY) of S_v-ZIS/5.0MoSe₂.

Fig. R7 was presented in Fig. 4c of the revised manuscript, the following

description are also displayed in the revised manuscript on Line 16-23/Page 15 to Line 1-4/Page 16 and highlighted:

Besides, Fig. S5 shows the wavelength dependent hydrogen evolution efficiency of S_v-ZIS/MoSe₂, which was tested following the similar procedure of photocatalytic H₂ evolution, except that the band-pass filter was equipped to obtain monochromatic incident light ($\lambda = 380, 420, 500$ and 600 nm). The detailed test results and the light power of different monochromatic light are displaying in Table S5. Accordingly, the AQY of photocatalytic H₂ evolution over the S_v-ZIS/MoSe₂ photocatalyst can be calculated (the detailed calculation process is shown in the Supporting Information) and the action spectrum was displayed in Fig. 4c. As observed, the action spectrum of S_v-ZIS/MoSe₂ matches well with the UV-vis absorption spectra, besides, the AQY values of S_v-ZIS/MoSe₂ are about 93.08% (380 nm), 76.48% (420 nm), 29.7% (500 nm) and 0.15% (600 nm), indicating the outstanding optical absorption and utilization capacity of S_v-ZIS/MoSe₂ photocatalyst.

Moreover, in the revised manuscript, according to the reviewer's suggestion, the AQY of ZIS and S_v-ZIS for H₂ evolution under 300 W Xe light equipping with different bandpass filter ($\lambda=380, 420, 500$ and 600 nm) were also tested and calculated. The action spectrum was presenting in **Fig. R8**. It can be observed that both the ZIS and S_v-ZIS present the action spectrum of photocatalytic H₂ evolution coinciding with the absorption spectrum. Moreover, in different monochromatic light wavelength, the AQY of S_v-ZIS are larger than that of ZIS, suggesting the more efficient photons to H₂ conversion ability of S_v-ZIS, which should be caused by the enhanced light absorption and the promoted photocarriers separation efficiency by introducing abundant S-vacancies in S_v-ZIS. Thanks again for providing us this valuable suggestion.

Fig. R8. Wavelength-dependent AQY for photocatalytic H₂ evolution of ZIS and S_v-ZIS.

Fig. R8 has been added into the Supporting Information of the revised manuscript and named as Fig. S6. The following description has also been added into the revised manuscript on Line 4-8/Page 16 and highlighted:

Fig. S6 is the AQY of ZIS and S_v-ZIS, it can be observed that under different monochromatic light wavelength, the AQY of S_v-ZIS are larger than that of ZIS, suggesting the more efficient photons to H₂ conversion ability of S_v-ZIS, which should be caused by the enhanced light absorption and the promoted photocarriers separation efficiency by introducing abundant S-vacancies in S_v-ZIS.

REVIEWERS' COMMENTS

Reviewer #1 (Remarks to the Author):

The manuscript has been improved well, and now can be accepted for publication.

Reviewer #2 (Remarks to the Author):

In my opinion the manuscript could be published in the current version. Authors have included reviewers' suggestions and improved the manuscript.

Reviewer #3 (Remarks to the Author):

Wang et al. have done an excellent analysis and thorough discussion based on the comments given by myself and the other reviewers. Although the overall quality of the manuscript has a marked improvement, I still do not recommend the manuscript to be published in the high-impact Nature Communications due to lack of novelty and scientific insights/merits of the overall ZnIn₂S₄/MoSe₂ system to the NC readers.

Furthermore, the reported performance for ZnIn₂S₄ is already rather high (at 3.36 mmol g⁻¹ h⁻¹), even compared to some of the already optimized systems of other works (specifically the ZnIn₂S₄/MoSe₂ systems listed in Table S6). Therefore, all this is to say that the high performance of the present Sv-ZnIn₂S₄/MoSe₂ as reported in this manuscript could also be influenced by what have already been reported in the literature, making the obtained high performance of 63.21 mmol g⁻¹ h⁻¹ from the Z-scheme system less impressive when putting into this context/perspective.

Since it is important to showcase the scientific merits to the readers in this pacey field, the use of computational analysis is of vital importance to be performed alongside the experimental analysis to further support the underlying fundamental understanding of the catalytic mechanism. For example, the magnitude and scale of electron distribution and charge transfer on the Mo-S bond could be validated on a theoretical level. As it stands, the authors only verified the existence of the Mo-S bond and hypothesized its synergistic function, but the degree at which this aids in the photocatalytic performance is not clear and very ambiguous to the NC readers. Therefore, this paper should be rejected from the consideration in Nature Communications, but can be aimed for the submission to Communications Chemistry upon thorough revision from the authors.

Response to editor

Dear Dr. Adam Weingarten,

Thanks for your letter for informing us that our manuscript entitled “**Interfacial Chemical Bond and Internal Electric Field Modulated Z-Scheme S_v-ZnIn₂S₄/MoSe₂ Photocatalyst for Efficient Hydrogen Evolution**” (Manuscript ID: NCOMMS-21-04662B) could be publish on Nature Communications in a suitably revised version, and also thanks for the reviewers’ work in reviewing our article. Herein, we have carefully studied the comments from reviewers’ comments and tried our best to revise the manuscript accordingly. All the changes or revisions of the manuscript have been highlighted by yellow color in the revised manuscript, which has also been presented in the following “**Point-by-point Response to the Reviewers’ Comments**” section. We sincerely hope that this reply will satisfy the reviewer, and deeply appreciate your consideration of our manuscript. If you have any queries, please don’t hesitate to contact us.

Best wishes,

Prof. Zhenjiang Li

Qingdao University of Science and Technology,

Qingdao, China.

E-Mail: zhenjiangli@qust.edu.cn

Point-by-point Response to the Reviewers' Comments

Response to Reviewer #1

Comments

The manuscript has been improved well, and now can be accepted for publication.

Response:

Many thanks for the reviewer's encouragement.

Response to Reviewer #2

Comments

In my opinion the manuscript could be published in the current version. Authors have included reviewers' suggestions and improved the manuscript.

Response:

We thank the reviewer for reviewing and providing insightful comments to improve the manuscript's quality.

Response to Reviewer #3

Comments

Wang et al. have done an excellent analysis and thorough discussion based on the comments given by myself and the other reviewers. Although the overall quality of the manuscript has a marked improvement, I still do not recommend the manuscript to be published in the high-impact Nature Communications due to lack of novelty and

scientific insights/merits of the overall ZnIn₂S₄/MoSe₂ system to the NC readers.

Furthermore, the reported performance for ZnIn₂S₄ is already rather high (at 3.36 mmol·g⁻¹·h⁻¹), even compared to some of the already optimized systems of other works (specifically the ZnIn₂S₄MoSe₂ systems listed in Table S6). Therefore, all this is to say that the high performance of the present S_v-ZnIn₂S₄MoSe₂ as reported in this manuscript could also be influenced by what have already been reported in the literature, making the obtained high performance of 63.21 mmol·g⁻¹·h⁻¹ from the Z-scheme system less impressive when putting into this context/perspective.

Since it is important to showcase the scientific merits to the readers in this pacey field, the use of computational analysis is of vital importance to be performed alongside the experimental analysis to further support the underlying fundamental understanding of the catalytic mechanism. For example, the magnitude and scale of electron distribution and charge transfer on the Mo-S bond could be validated on a theoretical level. As it stands, the authors only verified the existence of the Mo-S bond and hypothesized its synergistic function, but the degree at which these aids in the photocatalytic performance is not clear and very ambiguous to the NC readers. Therefore, this paper should be rejected from the consideration in Nature Communications, but can be aimed for the submission to Communications Chemistry upon thorough revision from the authors.

Response:

Many thanks for the reviewer's professional comment. However, as for the reviewer's comments, there are several issues we want to make further clarification.

Firstly, as for the novelty of our Z-Scheme $\text{ZnIn}_2\text{S}_4/\text{MoSe}_2$ photocatalytic system, it is true that both the $\text{ZnIn}_2\text{S}_4/\text{MoSe}_2$ system and Z-scheme photocatalyst have been reported by other researchers, however, through ingenious process design to construct interfacial chemical bond and internal electric field for consciously modulating Z-scheme charge transfer has been rarely reported, and it is important for realizing efficient photocatalytic property and revealing the photocatalytic reaction mechanism. In our work, the $\text{ZnIn}_2\text{S}_4/\text{MoSe}_2$ system was selected as the model photocatalyst due to the matching energy band and crystal structure (both ZnIn_2S_4 and MoSe_2 possess hexagonal layered crystal structure) between ZnIn_2S_4 and MoSe_2 . In the hydrothermal reaction involving hydrazine hydrate, the S vacancies and coordinative unsaturation S atoms would be generated on the surface of ZnIn_2S_4 , where the S vacancies can enhance light absorption and facilitate photocarriers separation, more importantly, the abundant coordinative unsaturation S atoms can serve as anchoring sites for Mo atoms, thus contributing to the in-situ growth of MoSe_2 on the surface of $\text{S}_v\text{-ZnIn}_2\text{S}_4$ by Mo-S bond. Through the above process, $\text{S}_v\text{-ZnIn}_2\text{S}_4/\text{MoSe}_2$ heterostructure connected by Mo-S bond can be fabricated. Due to the different Fermi level between $\text{S}_v\text{-ZnIn}_2\text{S}_4$ and MoSe_2 , an internal electric field at the interface of $\text{S}_v\text{-ZIS}$ to MoSe_2 would be built, which provided the necessary driving force to realize Z-scheme charge transfer inside $\text{S}_v\text{-ZnIn}_2\text{S}_4/\text{MoSe}_2$. The Mo-S bond can act as atomic-level interfacial “bridge” for promoting the photoexcited carriers migration between $\text{S}_v\text{-ZIS}$ and MoSe_2 , and accelerating the Z-scheme charge transfer inside $\text{S}_v\text{-ZnIn}_2\text{S}_4/\text{MoSe}_2$, which was furtherly confirmed by Surface photovoltage, DMPO spin-trapping EPR spectra and

the supplemented density functional theory (DFT) calculations (**Fig. R1**). Under the co-action of interfacial chemical bond and internal electric field, the Z-Scheme charge transfer mechanism was realized, and finally leading to the excellent photocatalytic H₂ evolution activity of S_v-ZnIn₂S₄/MoSe₂.

In conclusion, the novelty of our work not only reflects on developing particular defect-induced heterostructure constructing strategy, but also reflects on revealing the synergistic promoting mechanism of interfacial chemical bond and internal electric field on photocatalytic water splitting for H₂ evolution.

Secondly, due to the different preparation conditions adopted by different groups, the H₂ evolution efficiency for ZnIn₂S₄ would be different. For example, Lou's group reported that the photocatalytic H₂ evolution rate of ZnIn₂S₄ was 2.16 mmol·h⁻¹·g⁻¹ (*J. Am. Chem. Soc.* 2018, 140, 45, 15145–15148). The ZnIn₂S₄ photocatalyst synthesized by Yang's group exhibited a H₂ evolution rate of 6.42 mmol·h⁻¹·g⁻¹ (*Nano Energy* 76 (2020) 105031). Liu's group reported that the H₂ evolution rate over ZnIn₂S₄ was 8.42 mmol·h⁻¹·g⁻¹ (*J. Phys. Chem. C* 2011, 115, 13, 6149–6155). In this context, it is not strange for the two ZnIn₂S₄/MoSe₂ photocatalysts listed in Table S6 performed an inferior H₂ production activity than the ZnIn₂S₄ in our work. In addition, it should be noted that the H₂ evolution rate of S_v-ZnIn₂S₄/MoSe₂ in our work is almost 20 times higher than that of the original ZnIn₂S₄, which is also obviously greater than that in Ref. S10 and S17 in Table S6 (3.69 and 2.2 times respectively). Therefore, the efficient photocatalytic H₂ evolution rate of S_v-ZnIn₂S₄/MoSe₂ in our work was not realized based on the higher H₂ production rate of ZnIn₂S₄, but realized by the direct

Z-scheme charge transfer mechanism regulated by the atomic-level interface contact (Mo-S) and internal electric field.

Furtherly, just as the reviewer mentioned, it is important to reveal the photocatalytic mechanism by combining experimental analysis with theoretical simulation, therefore, for this revision, the density functional theory (DFT) calculations of $S_v\text{-ZnIn}_2\text{S}_4/\text{MoSe}_2$ were conducted out. **Fig. R1(a)** is the optimized structure of $S_v\text{-ZnIn}_2\text{S}_4/\text{MoSe}_2$ heterostructure, where the coordinative unsaturation S atoms was simulated by breaking two Zn-S bonds in the surface of ZnIn_2S_4 . According to Population analysis and Hirshfeld analysis results, the population of $\text{Mo}_{001}\text{-S}_{018}$ is 0.34, and the transferred charge between MoSe_2 and $S_v\text{-ZnIn}_2\text{S}_4$ is $0.12|e|$. The above results directly demonstrate the intense bonding effect between the Mo atom in MoSe_2 and the coordinative unsaturation S atom in ZnIn_2S_4 . **Fig. R1(b)** shows the side view of charge density difference of $S_v\text{-ZnIn}_2\text{S}_4/\text{MoSe}_2$ heterostructure, where the red and blue iso-surfaces denote the accumulation and depletion of electron density, respectively. As observed, the electron cloud density presents distinctly localized distribution between the Mo atom in MoSe_2 and the coordinative unsaturation S atoms in $S_v\text{-ZnIn}_2\text{S}_4$, which more intuitively manifests the intense bonding effect between Mo and S. Additionally, it can be noted that the surface of MoSe_2 was dominantly covered by red color, while $S_v\text{-ZnIn}_2\text{S}_4$ was chiefly filled by blue color, suggesting that the electrons in $S_v\text{-ZnIn}_2\text{S}_4$ were transfer to MoSe_2 along the intimate heterointerface, which would subsequently induce the internal electric field in $S_v\text{-ZnIn}_2\text{S}_4/\text{MoSe}_2$ heterostructure.

Fig. R1. (a) The optimized structure and (b) the side view of charge density difference of $S_v\text{-ZnIn}_2\text{S}_4/\text{MoSe}_2$ heterostructure.

Fig. R1 has been added into the revised manuscript and named as Fig. 7(a) and (b). The following description has also been added into the revised manuscript on Line 1-17/Page 24 and highlighted by yellow color.

To further reveal the photocatalytic reaction mechanism of $S_v\text{-ZnIn}_2\text{S}_4/\text{MoSe}_2$ heterostructure, the density functional theory (DFT) calculations were conducted out. Fig. 7(a) is the optimized structure of $S_v\text{-ZnIn}_2\text{S}_4/\text{MoSe}_2$ heterostructure, where the coordinative unsaturation S atoms was simulated by breaking two Zn-S bonds in the surface of ZnIn_2S_4 . According to Population analysis and Hirshfeld analysis results, the population of $\text{Mo}_{001}\text{-S}_{018}$ is 0.34, and the transferred charge between MoSe_2 and $S_v\text{-ZnIn}_2\text{S}_4$ is $0.12|e|$. The above results directly demonstrate the intense bonding effect between the Mo atom in MoSe_2 and the coordinative unsaturation S atom in ZnIn_2S_4 . Fig. 7(b) shows the side view of charge density difference of

S_v -ZnIn₂S₄/MoSe₂, where the red and blue iso-surfaces denote the accumulation and depletion of electron density, respectively. As observed, the electron cloud density presents distinctly localized distribution between the Mo atom in MoSe₂ and the coordinative unsaturation S atoms in S_v -ZnIn₂S₄, which more intuitively manifests the intense bonding effect between Mo and S. Additionally, it can be noted that the surface of MoSe₂ was dominantly covered by red color, while S_v -ZnIn₂S₄ was chiefly filled by blue color, suggesting that the electrons in S_v -ZnIn₂S₄ were transfer to MoSe₂ along the intimate heterointerface, which would subsequently induce the internal electric field in S_v -ZnIn₂S₄/MoSe₂ heterostructure.

The following content has also been added into the Experimental section on Line 12-20 /Page 30 of the revised manuscript.

Theoretical Calculation

Density functional theory (DFT) calculations were performed utilizing the CASTEP module of Materials Studio 6.1⁵⁵, the Perdew-Burke-Emzerhof (PBE) functional⁵⁶, and ultrasoft pseudopotential (USPP) method^{57, 58}. The cut-off kinetic energy of 400 eV, a 3×3×3 Monkhorst-pack k-point (Γ point) mesh sampled the Brillouin zone with a smearing broadening of 0.05 eV were applied during the whole process. The convergence criteria of self-consistent field (SCF), total energy difference, maximum force, and maximum displacement are 2.0×10^{-6} eV/atom, 2.0×10^{-5} eV/atom, 5.0×10^{-2} eV/Å, and 2.0×10^{-3} Å, respectively.

The above-mentioned references have been listed in the reference list of the revised manuscript.

55. Segall, M. D. et al. First-principles simulation: ideas, illustrations and the CASTEP code. *J. Phys.: Condens. Matter* **14**, 2717-2744 (2002).

56. Perdew, J. Burke, K. Ernzerhof, M. Generalized Gradient Approximation Made Simple. *Phys. Rev. Lett.* **77**, 3865-3868 (1996).

57. Vanderbilt, D. Soft self-consistent pseudopotentials in a generalized eigenvalue formalism. *Phys. Rev. B.* **41**, 7892-7895 (1990).

58. Perdew, J. Burke, K. Ernzerhof, M. Generalized gradient approximation made simple [Phys. Rev. Lett. 77, 3865 (1996)]. *Phys. Rev. Lett.* **78**, 1396-1396 (1997).

In conclusion, our manuscript not only presents obvious innovation on photocatalyst fabricating strategy, heterojunction combining mode and photocatalytic H₂ evolution activity, but also, more importantly, gives rise an advanced model for consciously realizing direct Z-scheme charge by synergistically modulating the interfacial atomic-level contact mode and internal electric field. Based on the above advantages, we believe that our paper should be published in the high-impact Nature Communications. Once it's published, it would afford a meaningful reference for Nature Communications' reader on constructing surface defects-induced heterostructure with interfacial chemical bond and intense internal electric field.

Thanks again for your professional comments.